# Uncalled4 improves nanopore DNA and RNA modification detection via fast and accurate signal alignment

Sam Kovaka [1] ✉, Paul W. Hook [2], Katharine M. Jenike[3], Vikram Shivakumar[1], Luke B. Morina[2], Roham Razaghi[2], Winston Timp [2,3] & Michael C. Schatz [1,3,4]

Nanopore signal analysis enables detection of nucleotide modifications from native DNA and RNA sequencing, providing both accurate genetic or transcriptomic and epigenetic information without additional library preparation. At present, only a limited set of modifications can be directly basecalled (for example, 5-methylcytosine), while most others require exploratory methods that often begin with alignment of nanopore signal to a nucleotide reference. We present Uncalled4, a toolkit for nanopore signal alignment, analysis and visualization. Uncalled4 features an efficient banded signal alignment algorithm, BAM signal alignment file format, statistics for comparing signal alignment methods and a reproducible de novo training method for *k*-mer-based pore models, revealing potential errors in Oxford Nanopore Technologies' state-of-the-art DNA model. We apply Uncalled4 to RNA 6-methyladenine (m6A) detection in seven human cell lines, identifying 26% more modifications than Nanopolish using m6Anet, including in several genes where m6A has known implications in cancer. Uncalled4 is available open source at github.com/skovaka/uncalled4.

Long-read single-molecule sequencers from Oxford Nanopore Technologies (ONT) and Pacific Biosciences (PacBio) have increasing utility in generating complete genomes and transcriptomes by improving resolution of complex DNA and RNA sequences[1–3]. These sequencers can also detect nucleotide modifications without any specialized library preparation, enabling genome-wide epigenetic profiling including within highly repetitive regions that could not be accurately aligned to with short reads[4]. Nanopore sequencing is unique in not relying on sequencing-by-synthesis, instead measuring electric current that varies over time as nucleotides pass through a pore. While many analyses only use the basecalled sequence, inclusion of the electric current can improve fidelity in several applications, including error correction[5,6], real-time targeted sequencing[7,8] and nucleotide modification detection[9]. Furthermore, ONT is currently the only commercially available platform for directly sequencing RNA without generation of complementary DNA (cDNA), enabling detection of epitranscriptomic modifications. Over 150 known RNA modifications are known to exist, although only a few can be comprehensively detected at the single-nucleotide level, with varying accuracy[10].

Early nanopore sequencers exhibited a high error rate, which could be improved via signal-based polishing[5] or advanced basecalling algorithms. However, a combination of improvements to sequencing chemistry and computational methods have decreased the average ONT DNA sequencing error rate to nearly 1%, making signal-based polishing largely unnecessary for DNA. This was achieved, in part, by a recent major DNA chemistry update to the r10.4.1 pore, which features two 'reader heads' rather than the one present in the previous standard, r9.4.1 (Fig. 1a). Direct RNA accuracy has lagged behind, where signal-based polishing can still improve splice site identification[6]. On the software side, modern basecallers use neural networks trained

[1]Department of Computer Science, Johns Hopkins University, Baltimore, MD, USA. [2]Department of Biomedical Engineering, Johns Hopkins University, Baltimore, MD, USA. [3]Department of Genetic Medicine, Johns Hopkins University, Baltimore, MD, USA. [4]Department of Biology, Johns Hopkins University, Baltimore, MD, USA. ✉e-mail: skovaka1@jhu.edu

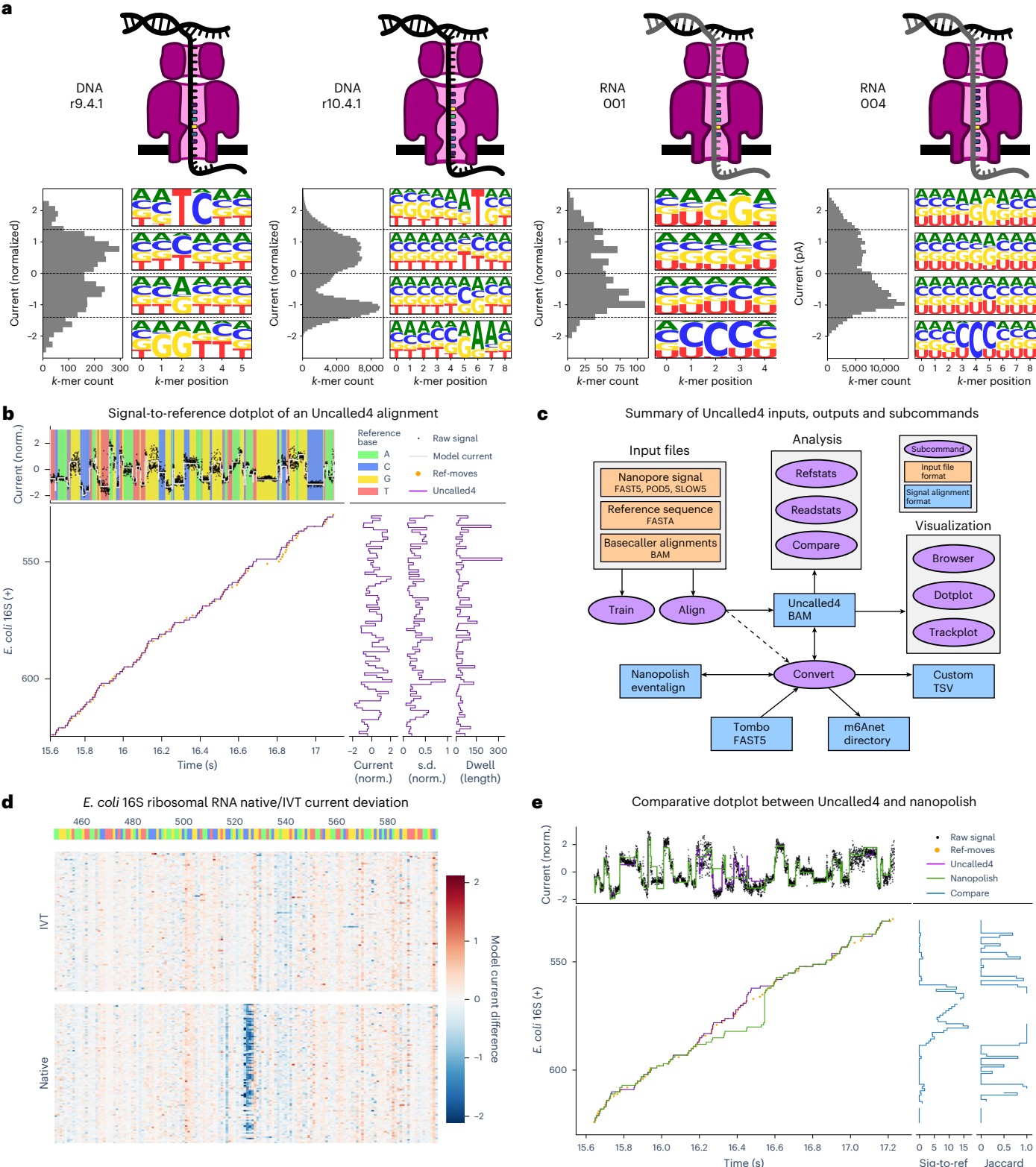

**Fig. 1 | Pore model and alignment methods overview. a**, Schematics of ONT sequencing chemistries, their pore *k*-mer model current distributions, and nucleotide compositions of *k*-mers within current ranges indicated by dashed lines. **b**, A signal-to-reference dotplot of an *Escherichia coli* 16S rRNA read sequenced using ONT r9.4 direct RNA sequencing. Top panel shows the raw samples (black) plotted over the reference base it was aligned to, with the expected pore model current in white. Main panel shows the Uncalled4 read alignment (purple line) over the projected basecaller metadata alignment (orange dots). Side panels show per-reference coordinate summary statistics for the alignment. **c**, Schematic of Uncalled4 inputs, outputs and subcommands

(Methods). **d**, A trackplot displaying heatmaps of many native (bottom) and IVT (top) *E. coli* 16S rRNA reads aligned by Uncalled4, colored by the difference between the observed and expected normalized current level. Top bar is colored by reference base, and an O6-methylguanine site is known to occur at position 526. **e**, A reflplot summarizing the distributions of differences between observed and expected normalized current levels for native (purple) and IVT (green) reads. **e**, A comparative signal-to-reference dotplot alongside distance (dist.) metrics between Uncalled4 and Nanopolish alignments of the same read, where line breaks in the distance plots correspond to regions masked by Nanopolish. norm., normalized.

on known sequences to translate the electrical signal into nucleotide reads, with the network architecture and training data being major factors in their accuracy[11]. However, ONT basecallers do not provide an accurate mapping between individual bases and the signal segments that represent them, instead requiring signal alignment to obtain such a mapping.

The latest ONT basecallers can directly detect 5-methylcytosine (5mC) and 5-hydroxymethylcytosine (5hmC) from individual DNA reads[12]. In humans, 5mC is the most common DNA modification, with wide-ranging effects in cellular development usually via suppression of transcription, with aberrant DNA methylation a frequent hallmark of several types of cancer[13,14]. Progress has also been reported from ONT on direct RNA basecalling for 6-methyladenine (m6A), the most common RNA modification in humans, using the early access RNA004 sequencing chemistry, currently limited to DRACH motifs (D = A, G or T; R = A or G; H = A, C or T) where over 60% of m6A sites occur in humans[15]. m6A is one of several known epitranscriptomic modifications, with diverse effects including transcript decay[16,17], transcript stabilization[18,19] and increased translational efficiency[20]. Beyond 5mC, 5hmC and m6A in limited contexts, detection of other modifications requires specialized computational methods run after basecalling. Some methods infer modifications from basecaller errors, but these are highly derived results that are sensitive to changes in basecalling models[10]. Instead, the most advanced methods analyze the raw signal directly, usually in combination with basecalled reads and beginning with alignment between the signal and a nucleotide reference.

Nanopore signal alignment (also known as event alignment[21,22], segmentation[9] and resquiggling[23]) is analogous to standard nucleotide alignment in variant calling, where the read aligner is often separate from the variant caller. This is also the case for many RNA modification callers that rely on Nanopolish[21] or Tombo[23], the two most commonly used signal aligners[10]. Signal alignment begins by translating the reference sequence into electrical current using a $k$-mer pore model, which maps each $k$-mer to the current expected when that combination of bases are in the pore. Processed read signal (for example, normalized) is then aligned to the expected reference current using a dynamic programming algorithm, such as dynamic time warping (DTW), as used by Tombo, or hidden Markov models, used by Nanopolish. The different methods generally produce similar, although slightly different alignments, with low-complexity sequences or noisy signal often resulting in large-scale disagreements. ONT basecallers can also output low-resolution signal alignments called 'moves', which approximately map blocks of 5–10 signal timepoints to basecalled read bases. Moves can be used in modification detection to approximately identify signal corresponding to a particular sequence, for example to extract signals representing 21–31 bases surrounding CpG sites[24,25], however, moves lack single-nucleotide resolution and thus require correction for more comprehensive analysis. Comparisons between signal alignment methods are hindered by the lack of a ground truth or a shared file format, causing most modification detectors to only support one aligner. More substantially, Nanopolish and Tombo have also not been updated for recent changes to ONT software and chemistry: neither supports the new r10.4.1 DNA or RNA004 chemistries or POD5 signal format, and Tombo relies on deprecated single FAST5 files. A more up-to-date alternative to Nanopolish is provided by f5c (ref. 22), a GPU implementation of the Nanopolish 'eventalign' algorithm with additional features including r10.4.1 DNA and RNA004 support, signal-to-read alignment, support for the open source SLOW5 format and an alternate BAM alignment encoding.

Here we present Uncalled4, a software toolkit for nanopore signal alignment, analysis and visualization (Fig. 1). Uncalled4 features command line tools and interactive visualizations for signal-to-reference alignments, methods for training new pore models, comparisons between alignment methods and modification level statistics. Uncalled4 uses a banded alignment algorithm guided by basecaller metadata, making it several times faster than Nanopolish or Tombo, and outputs an efficient and indexable BAM file that is directly convertible to widely supported and human-readable file formats. We use Uncalled4 to train a DNA pore model for the r10.4.1 pore, and apply this model to 5mC detection in CpG contexts (5mCpG). We also show Uncalled4 outperforms Nanopolish and Tombo in RNA modification detection using several different detection methods. We apply Uncalled4 and m6Anet to seven normal and cancer human cell lines and find 26% more sites supported by the m6A-Atlas than Nanopolish with equivalent precision, highlighting increased sensitivity in several genes with known m6A-related functions. Uncalled4 is implemented in C++ and Python, and is available open source at github.com/skovaka/uncalled4.

## Results

### Alignment efficiency, accuracy and visualization

Analogous to Nanopore read mapping, Nanopore signal alignment produces a mapping between stretches of nanopore electrical current and reference nucleotides. Conventional nucleotide alignments of basecalled reads (that is, from minimap2, ref. 26) determine the coordinates of the reference sequence, which is encoded into expected current using a $k$-mer pore model. Each pore model is specific to a particular sequencing chemistry, defined by molecule type (RNA or DNA), pore version (for example, r9.4.1, r10.4.1), sequencing speed (for example, 400 bases per second (bps)) and the output nucleotides, including possibly one or more modifications (for example, 5mCpG). The number of nucleotides affecting the current level varies by pore, with r10.4.1's double reader head necessitating longer $k$-mers than r9.4.1 and reduced noise in RNA004 enabling longer $k$-mers than RNA001 (Fig. 1a). In both DNA models, we find purines (A and G) at the central base enriched at low current levels and pyrimidines (C and T) enriched at high current levels, while the exact effect at each position varies between r9.4.1 and r10.4.1. RNA has a weaker high-level relationship between nucleotide content and current, with less consistency in which base has the dominant effect. We also note that all direct RNA pores (RNA001, RNA002 and RNA004) have highly similar current characteristics (Extended Data Fig. 1), although they differ in sequencing speed, yield and basecalling accuracy (Supplementary Note 1).

Uncalled4 uses basecaller-guided DTW (bcDTW) to rapidly and accurately align nanopore signals either to a reference genome and/or transcriptome (signal-to-reference) or to basecalled reads (signal-to-read) (Fig. 1b and Extended Data Fig. 2). While signal-to-read alignment is a promising method that avoids errors caused by genetic variation between the sample and the reference, most modification detection methods take advantage of the shared coordinate system and lack of basecaller errors provided by signal-to-reference alignment (Extended Data Fig. 3 and Supplementary Note 1).

Uncalled4 encodes signal alignments as per-reference-position statistics (that is current mean, current standard deviation and dwell time; Fig. 1b), which are efficiently stored in BAM tags alongside conventional nucleotide alignments. These BAM files can be either produced by Uncalled4's bcDTW algorithm via the align subcommand, or by Nanopolish, f5c or Tombo via the convert subcommand (Fig. 1c). We compare Uncalled4 with Nanopolish, Tombo and f5c alignments of DNA from *Drosophila melanogaster* (r9.4.1 and r10.4.1) and RNA from the human embryonic kidney 293T (HEK293T) cell line (RNA001 and RNA004). Uncalled is substantially faster than Tombo (2.9–6.8×), Nanopolish (1.7–1.9×) and f5c (1.3–2.7×, using a GPU with default parameters) (Table 1). The computational performance of Uncalled4 varies by molecule and pore type, and the file sizes grow linearly with the number of reads (Supplementary Fig. 1). The Uncalled4 compressed and indexable BAM format is over 20 times smaller than the Nanopolish or f5c raw eventalign format and six times smaller than eventalign with gzip compression (Table 1 and Supplementary Fig. 1). Uncalled4 also supports read signal input from FAST5, SLOW5/BLOW5 (ref. 27)

**Table 1 | Alignment time, disk space usage and distance from projected basecaller alignments for Uncalled4 and comparable Nanopore signal aligners**

| Sequencing chemistry | Signal aligner | File size (MB) | Time | Percentage of sites masked (%) | Model MAD | Median ref-moves distance | |
|---|---|---|---|---|---|---|---|
| | | | | | | Sig-to-ref | Jaccard |
| DNA r9.4 | Uncalled4 | **131.04** | **350.6** | **3.36** | 0.0908 | **0.3750** | 0.6000 |
| | f5c | 3,212.52 | 530.0 | 14.03 | **0.0865** | **0.3750** | **0.5833** |
| | Nanopolish | 3,212.52 | 653.8 | 14.03 | **0.0865** | **0.3750** | **0.5833** |
| | Tombo | 389.45 / 640.26 | 1015.0 | 23.66 | 0.2899 | 0.7000 | 0.8148 |
| DNA r10.4 | Uncalled4 | **132.25** | **244.7** | **2.62** | 0.1017 | **0.6429** | **0.7500** |
| | f5c | 3,706.41 | 574.4 | 10.54 | **0.0977** | **0.6429** | **0.7500** |
| RNA001 | Uncalled4 | **31.97** | **113.9** | **1.27** | 0.1355 | 0.5714 | 0.7222 |
| | f5c | 737.39 | 144.7 | 9.29 | 0.1369 | **0.5333** | **0.7143** |
| | Nanopolish | 737.50 | 194.9 | 9.29 | 0.1369 | **0.5333** | **0.7143** |
| | Tombo | 95.07 / 254.95 | 772.8 | 1.69 | **0.1317** | 0.6897 | 0.8182 |
| RNA004 | Uncalled4 | **32.91** | **63.7** | **1.96** | 0.0937 | 0.2500 | **0.5000** |
| | f5c | 541.68 | 70.6 | 9.50 | 0.1107 | **0.1304** | 0.5500 |

Tombo stores signal alignments alongside raw signal data in single FAST5 files, so space usage is reported as 'additional space'/'total FAST5 size'. 'Percentage of sites masked' indicates the fraction of per-read reference positions that were covered by basecalled read alignments but not by the signal aligner. 'Model MAD' is the MAD between per-read *k*-mer current means and the pore model. Sig, signal. Bold indicates the best performing method in each category.

and POD5 files, the last of which is the new ONT standard and not supported by Nanopolish, Tombo or f5c.

The median absolute difference (MAD) between per-*k*-mer normalized mean read current and the pore model can be used as an approximate measure of alignment quality, where values closer to zero indicate a closer match between the read and reference (Extended Data Fig. 3a). Uncalled4 has the lowest MAD for RNA004, and is within 0.004 normalized units of the lowest MAD for RNA001, r9.4.1 DNA and r10.4.1 DNA (Table 1 and Supplementary Table 1). Nanopolish and f5c fail to output 9–14% of sites that are covered by basecalled alignments, mostly due to masking sites that match the model poorly, which reduces the model MAD but also reduces sensitivity around modifications that strongly affect the current (Extended Data Fig. 4). Uncalled4 and Tombo perform less site-level masking, with Uncalled4 only masking sites around large insertions or deletions (>10 nt by default), however, Tombo only outputs 76% of basecaller-covered r9.4.1 DNA sites, mostly due read-level filtering likely caused by low-complexity sequences. Model MAD is therefore a limited metric, since higher MAD should be tolerated to accommodate nucleotide modifications and other 'noisy' signals.

We also compare Uncalled4 to other signal aligners by measuring distances between alignment coordinates of pairs of alignment methods (Fig. 1e and Supplementary Table 2), or between each alignment method and the basecaller ref-moves (Table 1 and Supplementary Table 1). Uncalled4 alignments consistently average within one nucleotide of the ref-moves and consistently perform best in mean signal-to-reference distance due to a higher frequency of large-scale alignment errors in Nanopolish, Tombo and f5c alignments (Supplementary Fig. 2 and Supplementary Note 2).

### Read signal and pore model characteristics

The nucleotide composition of *k*-mers at different current levels (Fig. 1a) demonstrate a complex relationship between nucleotide position and current. This can be quantitatively summarized by computing its 'substitution profile': the average normalized current change observed by substituting each base at each position for every *k*-mer in the model (Fig. 2a). We call the *k*-mer position with the most influence on current level the 'central base', and observe generally less influence at positions further from the central base, with the exception of r10.4.1 where a secondary reader head generates a smaller, but consistent

secondary effect near the beginning of the *k*-mer. In contrast to the DNA models, the profile of the RNA001 model is highly similar to the central five bases of RNA004, suggesting that the two RNA pores are structurally similar (Fig. 1a).

The reported intent of r10.4.1's double reader head is better accuracy around homopolymers. Homopolymers longer than the span of the reader head register little-to-no change as the same bases repeatedly pass through the pore, generating a higher frequency of deletions (Fig. 2b). Supporting this intent, we find r10.4.1 has a lower frequency of deletions in homopolymers nine nucleotides or longer in the *Drosophila melanogaster* genome compared to r9.4.1 (Fig. 2c). The deletion rate varies depending on which nucleotide the homopolymer is composed of, more so than the overall difference between r9.4.1 and r10.4.1, and is highest in cytosine with 26% of reads containing a deletion.

The length of homopolymers longer than the span of the pore (for example, 9-mers) can be estimated based on the dwell time, which measures how long the homopolymer occupied the pore. Such estimates are complicated by the high variability of dwell time, where the standard deviation of per-*k*-mer dwell times (14 raw samples for r10.4.1) is larger than the median (eight raw samples for r10.4.1) (Fig. 2d). Dwell time is also affected by sequence identity, depending both on the sequence at the pore and upstream where the motor protein is bound[28]. The position-specific influence of sequence on dwell time can be quantified by computing the median dwell time for each 5-mer at each offset relative to the central pore position, then computing the standard deviation of median dwell times for each set of 5-mers at each offset (Fig. 2e). Dwell time is usually most affected by the sequence 11–13 bases upstream, likely due to interactions with the motor protein, except in r10.4.1 DNA where the effect is slightly stronger within the pore. Dwell time standard deviation correlates with the mean dwell time, with the slowest sequencing speed (RNA001 70 bps) yielding the highest overall standard deviation. Guanine (G) generally increases dwell time, while cytosine (C) generally decreases dwell time, although the effect depends on the position relative to the pore and motor protein (Supplementary Fig. 3). These dwell time effects likely influence basecalling, particularly in homopolymers where it is the primary feature that determines length.

Uncalled4 provides a method to iteratively train pore models by repeatedly aligning signal and averaging signal characteristics for each *k*-mer (Methods and Supplementary Note 3). We use this training

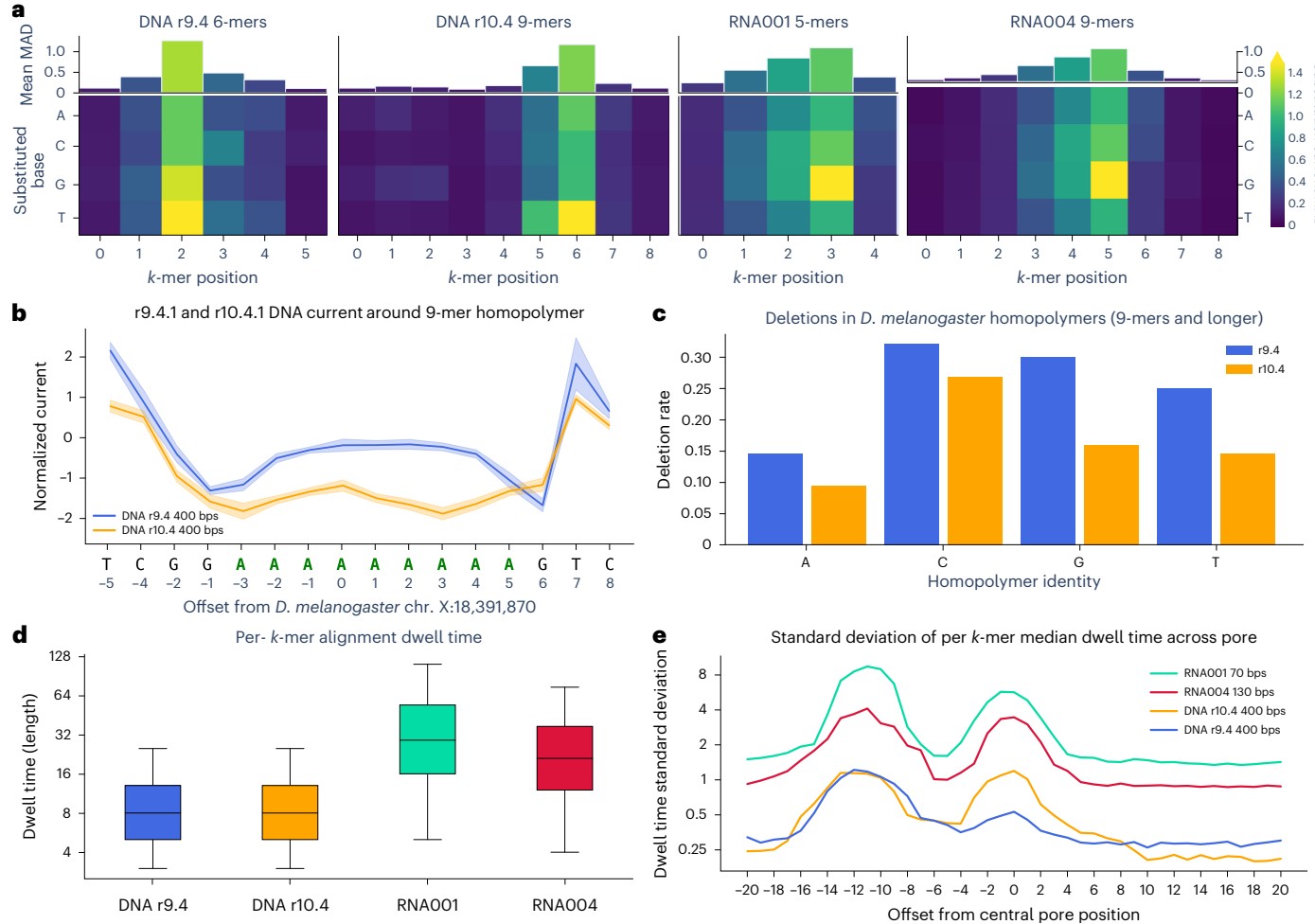

**Fig. 2 | Current distribution and nucleotide composition of *k*-mers in Uncalled4-trained pore models. a**, Pore model *k*-mer substitution profile heatmaps: the mean normalized current difference observed by substituting each base (*y* axis) at each *k*-mer position (*x* axis) averaged over all *k*-mers in the model. **b**, Mean and standard deviation of current surrounding a 9-mer adenine homopolymer in the *D. melanogaster* genome, based on Uncalled4 alignments of r9.4.1 and r10.4.1 DNA reads. Solid lines show the mean normalized current of all reads aligned at each position, and the shaded region shows the standard deviation of the mean current levels. **c**, Fraction of basecalled reads containing

a deletion within homopolymers of length nine or longer in the *D. melanogaster* X chromosome, computed using samtools mpileup. **d**, Distributions of per-read *k*-mer dwell times output by Uncalled4 alignments of 4,000 randomly sampled reads from *D. melanogaster* DNA and human HEK293T RNA. Boxes span the first and third quartiles with the median indicated by the horizontal line, and whiskers extend to 1.5 times the interquartile range. **e**, Standard deviation of per-*k*-mer dwell times relative to central pore position, where each offset along the *x* axis indicates the standard deviation of median dwell times for each 5-mer at that position.

procedure on PCR-amplified *D. melanogaster* DNA sequenced using r9.4.1 and r10.4.1 pores, showing strong agreement with corresponding pore models released by ONT (Extended Data Fig. 5). A notable set of outlier *k*-mers between the Uncalled4 and ONT r10.4.1 400-bps model have a consistent NNNNTVTTN motif (N, any base; V, not T), possibly caused by errors in the ONT model as evidenced by the *k*-mer substitution profile, comparison to ONT's r10.4.1 260-bps model and modification detection performance (Extended Data Fig. 5b–d and Supplementary Note 8). We also trained a r9.4.1 direct RNA (RNA002) model using in vitro transcribed (IVT) human HeLa cell line data, and the resulting model strongly correlates with both ONT's legacy 'rna_r9.4_180mv_70bps' and the RNA004 model, again demonstrating high similarity in current characteristics of these sequencing chemistries (Supplementary Note 3 and Extended Data Fig. 1b).

**DNA modification model training and detection**

To explore the effect of DNA modifications on the r10.4.1 DNA sequencing, we sequenced *D. melanogaster* DNA treated with CpG methyltransferase M.SssI to broadly modify CpG sites with 5mC (5mCpG), with

an average per-site methylation rate of 88% estimated by the Guppy basecaller (v.6.4.8, high-accuracy model). We trained a 9-mer model on the 5mCpG *D. melanogaster* dataset and compared the current levels to the unmodified model, confirming that *k*-mers with CpG in the central position were the most divergent (Fig. 3a and Supplementary Fig. 4a–c). In contrast, CpGs in the first five *k*-mer positions (secondary reader head) provide almost no information (Supplementary Fig. 4a). We also used Uncalled4 and f5c to directly detect 5mCpG methylation by comparing current levels between PCR and 5mCpG *D. melanogaster* r10.4.1 data using two-sample Kolmogorov–Smirnov (KS) test statistics or *z*-scores to compare current distributions surrounding CpG sites (Supplementary Note 4). The pattern of KS statistics and *z*-scores was highly similar for Uncalled4 and f5c, both showing a primary and secondary peak in each statistic, consistent with r10.4.1's double reader head (Fig. 3b and Supplementary Fig. 4d,e).

In addition to naturally occurring DNA and RNA modifications, Uncalled4 can train models including artificial modifications. We demonstrate this by training a pore model on *Saccharomyces cerevisiae* DNA with constitutive incorporation of BrdU, a thymine analog used to

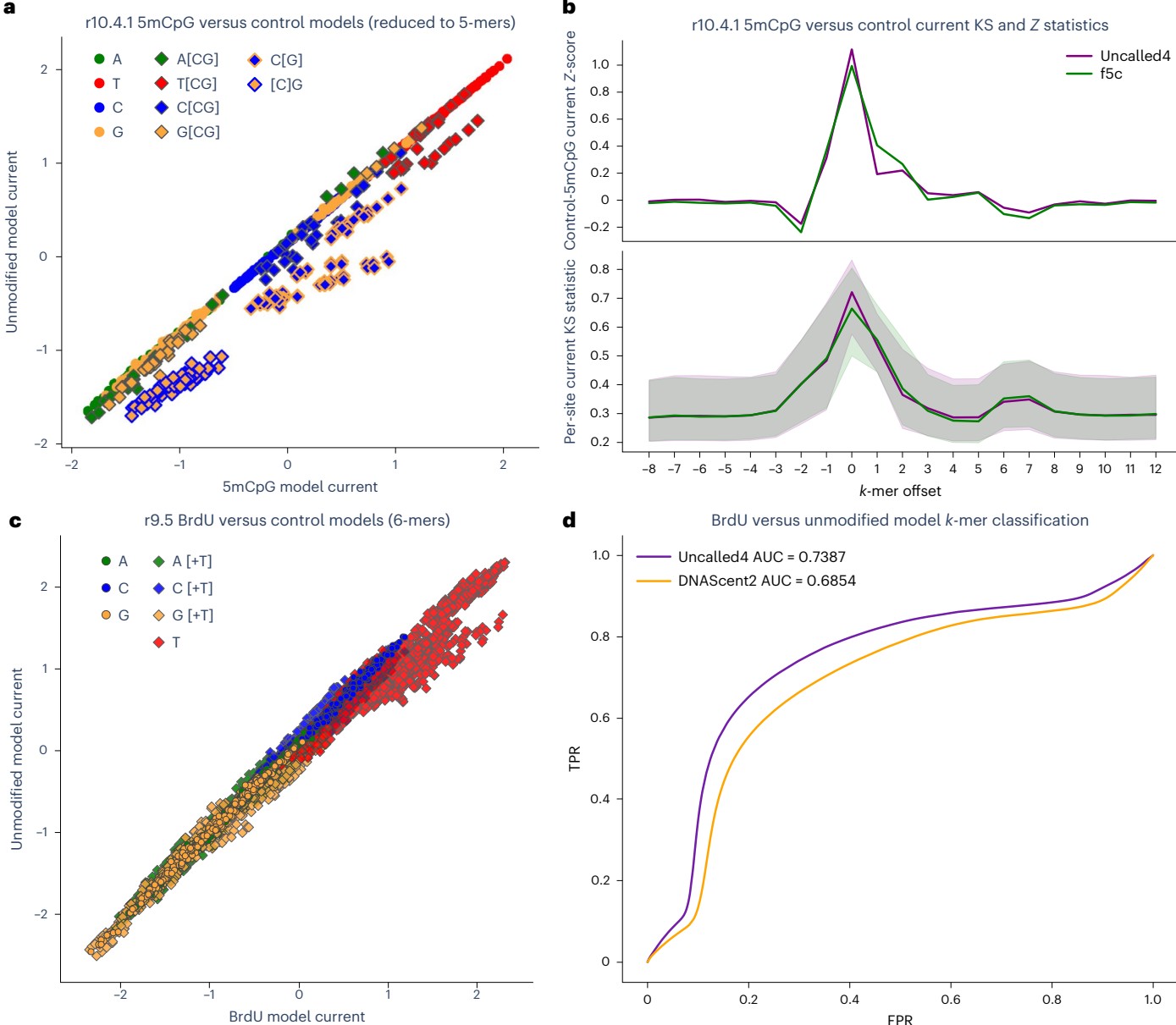

**Fig. 3 | 5mCpG signal characteristics. a**, Normalized current levels for Uncalled4 5mC (*x* axis) and unmodified control (*y* axis) r10.4.1 pore models, reduced to 4-mers by averaging *k*-mers sharing their last four bases. Each point is colored by the identity of the central base, with diamonds representing CpG containing *k*-mers. Outlined diamonds indicate *k*-mers with the modified cytosine at central position (C[G]) or one base upstream ([C]G). **b**, Current-level *z*-scores and KS statistics of differences in current between control and 5mCpG datasets for each position surrounding CpG sites in the *D. melanogaster* X chromosome, computed from Uncalled4 and f5c r10.4.1 signal alignments using the ONT r10.4.1 400-bps

model. KS statistic lines show the median normalized current difference between the signal aligned at each position and the pore model, and shaded regions show the interquartile range of the current difference. **c**, Normalized current levels for Uncalled4 BrdU and unmodified datasets trained on r9.5 data, with each point colored by the identity of the central position (second base) of each 5-mer and diamonds indicating *k*-mers containing a 'T'. **d**, ROC curve of classification of BrdU versuss unmodified *k*-mers based on Uncalled4 alignments using the Uncalled4-trained or DNAScent2 pore models compared to a control model.

label newly synthesized DNA[29] (Fig. 3c and Supplementary Fig. 5a,b). We compare this to a model trained on the same data using DNAscent[29] by aligning a mixture of BrdU-modified and unmodified reads using each BrdU pore model and a control DNA model, and find that the Uncalled4 model can better classify reads as BrdU by comparing current levels in *k*-mers with a single 'T' in their central position (Fig. 3d, Supplementary Fig. 5c and Supplementary Note 5). We acknowledge that DNAscent uses a more sophisticated classification approach that enables classification of *k*-mers with 'T's outside the central position; our analysis demonstrates that model training improves accuracy for Uncalled4 alignments using this simple statistic.

## Comparative RNA modification detection

To measure the effectiveness of Uncalled4, Nanopolish and Tombo in RNA modification detection, we begin with a comparative approach using two-sample KS statistics to measure dissimilarity of per-reference current distributions in two datasets with differing modification rates. We limit our analyses to RNA001 and RNA002, as there are currently far fewer publicly available RNA004 datasets and modification detection methods. We first use each aligner to compute KS statistics to detect a diverse set of 36 annotated *Escherichia coli* ribosomal RNA (rRNA) modifications by comparing native and IVT datasets, showing that Uncalled4 has consistently higher areas under the receiving-operator

curve (AUROC) and precision-recall curve (AUPRC) using 100× coverage at each site (Supplementary Fig. 6).

We next apply comparative methods to detect m6A using two human HEK293T cell line samples: wild-type and METTL3 m6A methyl-transferase knockout (KO). Accuracy was estimated using m6ACE-seq labels filtered for METTL3-sensitive sites[30]. Using transcript-level KS statistics, Uncalled4 outperforms Nanopolish and Tombo in AUPRC and AUROC, both in all contexts and limited to DRACH motifs (Fig. 4a and Extended Data Fig. 6). We also input the Uncalled4 and Nanopolish alignments into xPore[30], again yielding higher AUPRC and AUROC for Uncalled4 (Fig. 4a, Extended Data Fig. 6 and Supplementary Fig. 7). We additionally tested two methods for calling gene-level m6A sites: transcript-to-gene averaging and spliced genome alignment, the latter of which is a unique capability of Uncalled4 (Extended Data Fig. 7 and Supplementary Note 6). Uncalled4 performs best in both AUPRC and AUROC using either gene-level detection method (Fig. 4a and Extended Data Fig. 8).

## RNA modification detection with m6Anet

To detect RNA m6A sites without a matched control dataset we input Uncalled4 alignments to m6Anet (ref. [31]), a specialized m6A detection method designed for Nanopolish that uses a neural network to call m6A modifications at DRACH sites on individual reads before aggregating them to the transcript level. We retrained m6Anet using Uncalled4 alignments of the same HCT116 dataset that was used in the default m6Anet model for Nanopolish (Methods). We then compare m6Anet performance on a single wild-type HEK293T sample aligned with Uncalled4 or Nanopolish with previously published HEK293T GLORI labels[32] to estimate accuracy, demonstrating higher AUPRC and AUROC (Fig. 4b and Supplementary Fig. 8). Uncalled4 with m6Anet outputs 17% more candidate sites than Nanopolish due to less read- and site-level filtering, and Uncalled4 has consistently higher AUPRC when only including sites output by both aligners (intersection), by either aligner (union), or all sites sufficiently covered by basecalled alignments (cov ≥ 20×, Fig. 4b). Uncalled4 finds disproportionately more m6A sites in low-coverage regions (Fig. 4c and Supplementary Note 7), and the same trends are observed when sites are averaged to the gene level (Extended Data Fig. 9). We also used m6Anet to compare wild-type and METTL3-KO samples (Supplementary Note 7 and Supplementary Fig. 9), and found Uncalled4 + m6Anet outperforms KS statistics and xPore in DRACH contexts using any aligner (Fig. 4a and Extended Data Figs. 6 and 8).

We next compared the prevalence of m6A modifications using m6Anet in seven human cell lines, consisting of three normal and four cancer tissues (Fig. 5a). We used two replicates for most samples, except for the human mammary epithelial cell (HMEC) line where only one replicate was available, and HEK293T where three were used (Supplementary Table 3). We used m6A-Atlas v.2 as our set of putatively m6A-positive sites, noting that while this is an imperfect ground truth, the relative performance between different methods is similar whether GLORI or m6A-Atlas labels are used, showing it can be used to identify putative true positive (pTP) sites. (Supplementary Fig. 10 and Supplementary Note 8).

Beginning with transcript-level m6Anet calls on primary alignments, we find that Uncalled4 alignments yield consistently higher AUPRC and AUROC than Nanopolish on all samples (Supplementary Table 4). Uncalled4 also has generally higher recall and precision using the default probability cutoff of 0.9, with 18% more true positive m6A sites on average. The same patterns are observed when probabilities are averaged to the gene level, with more sites found by both aligners in every sample and Uncalled4 again finding 18% more true positives on average (Fig. 5a,b and Supplementary Table 5). Precision is lowest for NA12878 using either aligner, even more so than the other samples absent from m6A-atlas (HMEC and K562), likely because NA12878 has the lowest data quality as measured by observed yield, basecaller pass

and fail rate and quality scores (Supplementary Table 3). To correct the variable precision estimates, we adjusted the probability thresholds for each sample such that each has equal precision of either 80, 85 or 90% (Supplementary Table 4). Uncalled4 finds 27.3 and 28.1% more pTP sites on average at 80 and 85% precision respectively, but finds 0.1% fewer pTP sites at 90% precision. The lesser performance at 90% precision is likely due to the unreliability of m6A-Atlas labels at this threshold, where notably Uncalled4 finds more pTP sites in HEK293T, the most represented cell line in the m6A-Atlas. We therefore use the set of modifications found at 85% precision for further analysis, yielding higher recall than the default threshold in every cell line except NA12872 (Fig. 5a).

Uncalled4 finds disproportionately more m6A at sites at low coverage due to Nanopolish's pervasive masking (Fig. 5b). Sites found by either aligner are generally enriched around stop codons and in the 3' untranslated region, consistent with previous findings (Extended Data Fig. 10a,b). Most m6A sites that are only found by Uncalled4 are in the m6A-Atlas, and most sites not present in the m6A-Atlas are found by both Nanopolish and Uncalled4 (Fig. 5c). For both methods, approximately half of all sites were only identified in one sample, 30–32% of which are absent from the m6A-Atlas. The total number of m6A sites generally decreases with the number of supporting samples, as does the putative false positive rate, with the notable exception of sites shared by all seven samples, which is greater than those shared by only six samples (Extended Data Fig. 10c). This set of m6A sites shared by all samples indicates transcripts that are broadly modified; for example, the gene *c-Myc* has seven modifications found by Uncalled4 in all samples, where m6A has a well studied stabilizing effect on the c-Myc messenger RNA (mRNA)[18].

Aggregating the per-sample gene counts further, we compute the total number of modifications found across all samples for each gene by Uncalled4 and Nanopolish, revealing Uncalled4 broadly finds more modifications than Nanopolish (Supplementary Table 5). Specifically, we find more m6A in 66% of genes, fewer in 18% of genes and an equal number in 16% of genes (Fig. 5d). To further explore differences in m6A count in genes across the healthy and cancerous cell lines, we focus our analysis on COSMIC Census tier 1 genes[33], which identifies genes with mutations implicated in cancer development, and specifically only those with m6A present in all samples (Fig. 5e). Among this subset, the gene with the largest increase in m6A count between Uncalled4 and Nanopolish is *ABL1*, an oncogene that fuses with *BCR* in chronic myeloid leukemia (CML). *ABL1* has been identified as a potential target of the ALKBH5 demethylase[34], and it has been observed that m6A contributes to aberrant translation in BCR-ABL1 positive CML cases[35]. Incidentally, we find multiple m6A-containing reads that support the BCR-ABL1 fusion in the CML K562 cell line, demonstrating the long-range information provided by Nanopore sequencing (Fig. 5f). The gene with the next-highest increase in m6A count is the oncogene *JUN*, which is a known target of the METTL3 methyltransferase and its translation is promoted by m6A modification[20]. Several other genes in this subset are known to be transcriptionally destabilized by m6A: *STK11* (ref. [16]), *ID3* (ref. [36]), *AKT1*, *AKT2* (ref. [17]) and *NCOR2* (ref. [34]). Others are known to be stabilized by m6A, such as *c-MYC*[18] and *THRAP3* (ref. [19]).

Furthermore, among the top ten genes ranked by increase in m6A sites with Uncalled4 is *TTC4*, with a total 60 sites identified by Uncalled4 and 35 by Nanopolish across all cell lines, notably none of which are in the m6A-Atlas (Supplementary Table 5). However, closer inspection revealed that m6A-Atlas assigned all *TTC4* labels to the *MROH7-TTC4* readthrough transcript, which entirely contains *TTC4* (Fig. 5g). Most reads that align to *MROH7-TTC4* also multimap to *TTC4*, and *TTC4* m6A modification has been implicated in lung sepsis response in mice[37], suggesting that m6A-Atlas has mislabeled which gene the genomic coordinates should correspond to. *TTC4* contains a 34 amino acid tetratricopeptide repeat, which makes short-read alignment less reliable and may contribute to inaccuracies in its m6A-Atlas labels.

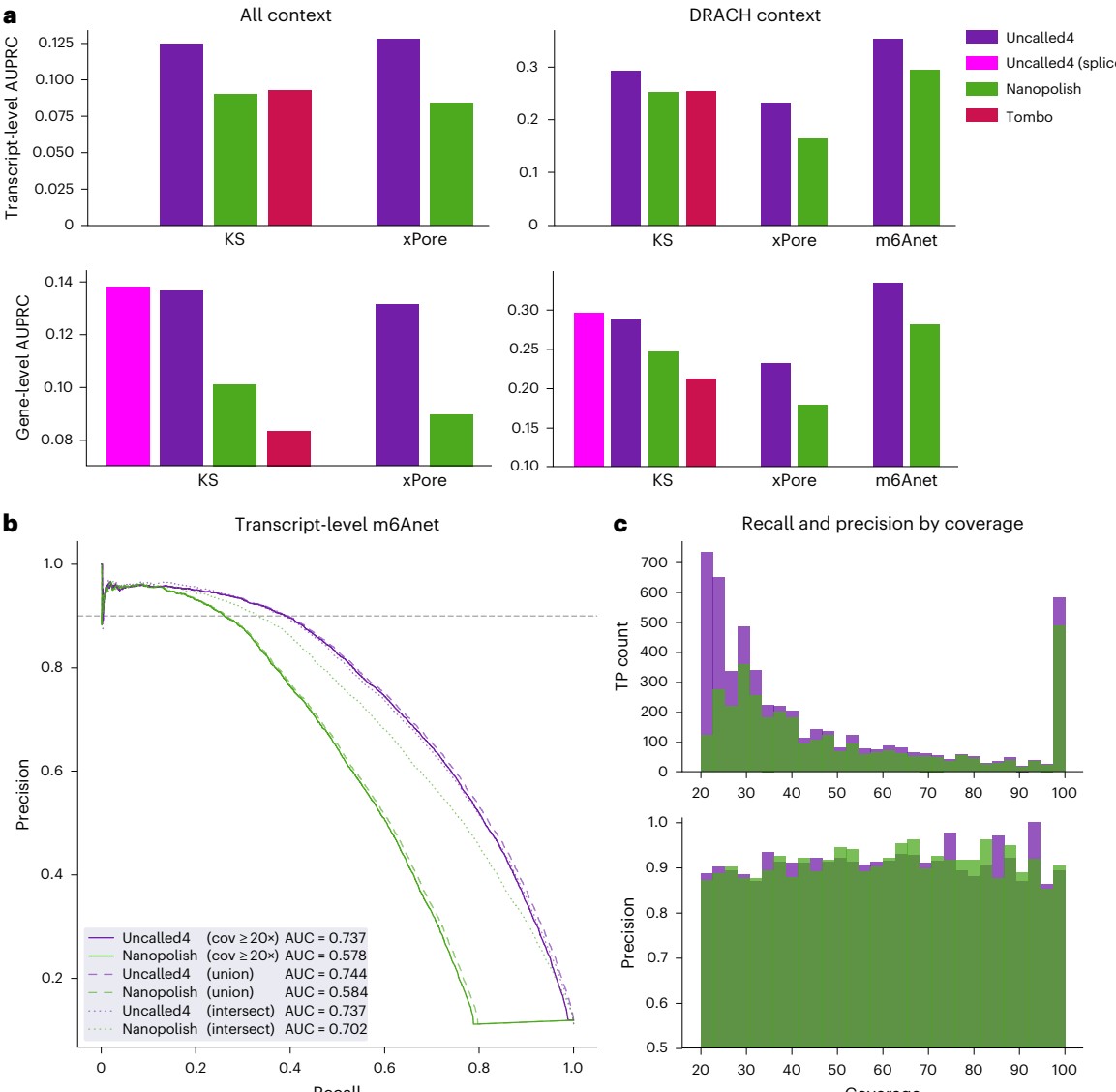

**Fig. 4 | HEK293T RNA m6A detection. a**, AUPRC for comparative RNA m6A modification detection using Uncalled4, Nanopolish and Tombo alignments as input to KS statistics, xPore or m6Anet. 'Uncalled4 (spliced)' (magenta) is based on spliced genome alignments, while all others use transcriptome alignments averaged to the gene level. **b**, Precision-recall curves comparing m6Anet performance on a single HEK293T sample using Uncalled4 or Nanopolish alignments. Solid lines include all sites that are covered by basecalled alignments by at least 20 reads, where sites not output by either tool are assigned a score of zero, generating a large discontinuity for Nanopolish due to pervasive masking. The dashed curves include all sites output by either aligner (union), while dotted curves only include sites output by both aligners (intersection). **c**, True positive rate and precision binned by basecalled read coverage. m6Anet probability threshold was selected such that the overall precision for each aligner equals 90% (dashed horizontal line in **b**).

## Discussion

Nanopore signal is information rich, encoding much more than the four canonical bases obtained from standard basecalling. Signal-to-nucleotide alignment is a critical step in extracting this information, but the process is error prone and few standards exist for comparing alignment methods. Uncalled4 features a rapid and highly accurate alignment algorithm guided by basecaller metadata, a compressed and indexed BAM-based signal alignment file format, and analyses to facilitate comparisons between signal alignment methods. Uncalled4's pore model training method is fully reproducible, requires no previous $k$-mer based model and reveals potential errors in ONT's official r10.4.1 DNA model. Accurate signal alignment enables more sensitive DNA and RNA modification detection than comparable signal aligners, enabling it to find substantially more RNA m6A sites in several disease-relevant genes using m6Anet in healthy and cancer human cell lines compared to Nanopolish.

A major benefit of epigenetic profiling with long reads is that the genetic identity is maintained, in contrast to short-read methods that involve base substitutions or read truncation, making it possible to comprehensively measure single-nucleotide, structural and epigenetic variation in one assay. In principle, these methods could be applied to a wide variety of samples with publicly available nanopore sequencing data, however, the raw signal required to identify modifications is often not made available, mostly due to large file sizes and lack of database support. Uncalled4's BAM format efficiently provides the statistics required by most signal-based detection methods in a widely supported format. A similar BAM tag was recently introduced by Squiguliser[38], a nanopore signal alignment visualizer in part inspired by an early version of Uncalled4, however, this only stores signal coordinates and not the current-level data required for modification detection. The use of efficient and indexable data representations will become even more critical as long-read sequencing becomes more widely adopted. In

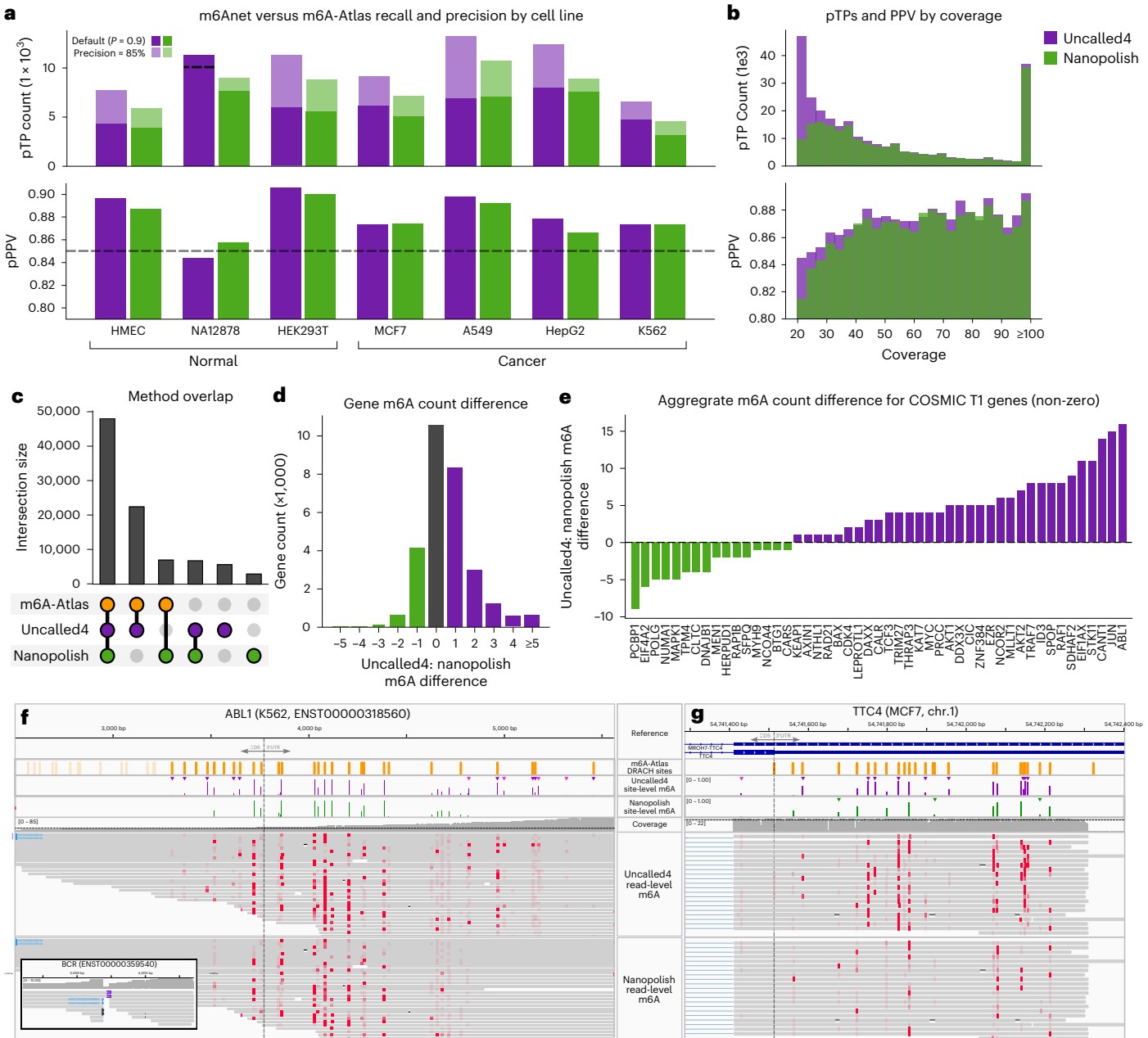

**Fig. 5 | RNA modification detection across seven human cell lines. a**, Number of m6A sites found in each cell line that occur in the m6A-Atlas v.2 pTPs. Solid bars indicate the number of sites found with the default probability threshold 0.9, and shaded bars indicate the count at threshold where the putative positive predictive values (pPPV) is 85%. Uncalled4 with NA12878 has reduced recall at 85% pPPV, as indicated by dashed line. **b**, Coverage distribution of true positive (pTP) sites (top) and pPPV of sites within coverage bins. **c**, Number of sites shared by Uncalled4, Nanopolish and m6A-atlas v.2 across all cell lines. **d**, Difference in per-gene m6A count found by Uncalled4 and Nanopolish across all seven cell lines. **e**, Difference in aggregated gene m6A count found by Uncalled4 versus Nanopolish alignments, limited to COSMIC tier 1 genes where at least one m6A modification is found in every cell line by either tool (51 genes). Negative (green) values indicate genes where more m6A sites were found by Nanopolish, and positive (purple) values indicate more m6A sites found by Uncalled4. **f**, Transcript-level m6A calls in an ABL1 transcript alongside BCR fusion. **g**, Gene-level m6A calls in the TTC4 gene.

addition to widespread clinical sequencing with long reads, the Human Pangenome Reference Consortium is using both ONT and PacBio sequencing to assemble a haplotype-resolved human pangenome[39]. Long-read pangenomes present the opportunity and challenge of pan-epigenomic analysis, complicated by every cell having a potentially unique and dynamic epigenome, and multiple types of nucleotide modification present across species. Uncalled4 provides a step toward scaling such analyses as more data becomes available.

Even more daunting than pan-epigenomics, a pan-epitranscriptomic catalog would need to account for the underlying dynamic nature of

the transcriptome and a much wider array of RNA modifications. The most well studied RNA modifications play an important role in RNA stability, mRNA splicing, mRNA export, translation efficiency and several other important roles[40]. The lack of training data is now the major factor preventing identification of most of the over 150 known RNA modifications. Certain modifications may also generate minute changes in signal, meaning accurate signal alignment is necessary to reveal these subtle changes. For modifications that can be detected but not identified for lack of accurate labels, Uncalled4's visualizations (Fig. 1) and analyses can serve as useful exploratory tools.

Our work showed that aggregating transcript-level m6A calls to the gene level is a straightforward approach to improve accuracy, however, this eliminates potentially interesting transcript-specific results. Detailed exploration of transcript-level modifications cannot rely solely on labels from short-read assays, which generally do not provide transcript-level specificity. Long-read methods must also be improved to accurately assign reads to transcripts in multi-isoform genes. Conventional transcriptome alignment often maps reads to incorrect transcripts by trimming alternatively spliced regions, or fails to include all potential mappings of fragmented reads, as shown here and in previous work[41]. Spliced genome alignment is an alternative approach that avoids isoform alignment ambiguity, but fully interpreting genome alignments would require mapping and disambiguating reads from the genome to transcriptome, similar to reference-guided transcriptome assembly[42]. If transcript-level modifications could be accurately identified, such methods could be applied to allele-specific modifications, similar to recent work in conventional transcriptomics[2]. Long reads are also well-suited for characterizing unannotated transcripts, or noncanonical transcripts generated by structural variation or circular RNAs, the last of which are associated with RNA modifications such as m6A (ref. 43).

We have presented a toolkit for nanopore signal alignment and analysis, focusing on applications in nucleotide modification detection. Signal alignment is useful in other applications, for example in several recent rapid signal mappers designed for targeted sequencing[7,8,44,45]. Uncalled4 could be useful in optimization of such approaches, and the Python module is already used by Sigmoni for basic signal processing[44]. Uncalled4 will also be valuable in understanding future updates to nanopore sequencing chemistry, and will aid other signal-based methods in adapting to those changes.

## Online content

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

## Methods

### Uncalled4 overview

Uncalled4 is a Python module and command line utility, with many computationally intensive subroutines implemented in C++ with Python bindings provided via PyBind11. The command line functionalities are split into several subcommands (Fig. 1f): 'align' implements the basecaller-guided signal alignment algorithm, which outputs a BAM file by default: 'convert' converts between signal alignment formats, where BAM and eventalign formats support input and output, Tombo FAST5s only support input, and m6Anet and TSV files only support output; and 'train' iteratively applies the alignment algorithm to train pore models and outputs a directory with $k$-mer models produced in each iteration. The remaining commands are divided into analysis and visualization. The analysis commands are refstats, which outputs reference-level statistics (for example, KS statistics), readstats, which outputs read-level statistics (for example, mean normalized model difference) and compare, which compares two BAM files containing the same set of reads aligned using difference methods (for example, Uncalled4 and Nanopolish). Visualization commands display interactive Plotly visualization, either as HTML files exportable to SVG or PNG, or as web browser sessions: dotplot displays one or more alignments of a signal read (Fig. 1b,c), trackplot displays one or more alignment tracks of many reads aligned to a reference region (Fig. 1d) and the browser command runs a local Dash web server that displays an interactive alignment track that can be clicked to display summary statistics, a dotplot and per-reference statistics distributions (Fig. 1e).

The align command is described in the following 'Signal preprocessing' and 'Basecaller-guided DTW' sections, convert in the 'Alignment encoding and formats' section, train in 'Pore model training', analysis commands in 'Analysis of signal alignments' and visualization commands in 'Visualizations'.

### Signal preprocessing

Before alignment, the raw nanopore electrical signal must be preprocessed to reduce noise and correct for systematic bias in the current levels. First, the individual sensor readings (raw samples) are segmented into 'events' using the same algorithm as UNCALLED[8], which uses rolling $t$-tests to group samples with similar current levels. This groups signal representing the same nucleotides, although variable sequencing speeds result in frequent 'stays' (consecutive events representing the same $k$-mer, roughly 50% of events) and fewer frequent 'skips' (an event representing multiple $k$-mers, ~1–5% of events). These events are stored with their sample start, length (proportional to dwell time), current mean and current standard deviation. Event detection parameters are chosen depending on the sequencing chemistry, where RNA uses longer $t$-test window lengths than DNA to adjust for the slower sequencing speed.

After event detection, the event current means are iteratively normalized to correct for systematic deviation from the pore model. Each iteration performs a linear transformation of the read signal defined by a multiplicative scale factor and an additive shift. The first iteration transforms the event mean currents ($E$) such that their distribution has the same mean and variance as the pore model $k$-mer mean currents within the reference coordinates ($K$) indicated by the basecaller alignments: scale = $\sigma(K)/\sigma(E)$; shift = $\mu(K) - \text{scale} \times \mu(E)$ (where $\sigma$ is the standard deviation and $\mu$ is the mean). This is similar to the 'method-of-moments' widely used as a simple normalization method[8,46,47], but using the reference sequence rather than assuming a random $k$-mer distribution. The second iteration performs linear regression between the aligned current means and the corresponding reference $k$-mer model current, where scale is the output slope and shift is the intercept, importantly averaging all raw samples aligning to each $k$-mer such that each $k$-mer contributes to the regression equally regardless of dwell time. This second iteration is not performed in the train subcommand by default to avoid over-fitting to an error-prone model. The Theil–Sen estimator, a nonparametric regression algorithm used by Tombo, was also tested, but this was found to be less accurate and slower than simple linear regression.

### bgDTW

Uncalled4 uses DTW to align preprocessed signal to a reference sequence guided by basecalled read alignments. DTW is a widely used dynamic programming algorithm that has previously been applied to nanopore sequencing by Tombo and others[23,46]. Nanopolish uses a hidden Markov model for alignment, which uses more complex transition probabilities that are trained on real data, but is otherwise similar to DTW in time and space complexity. The most basic form of DTW has $O(N \times M)$ complexity, where $N$ is the number of read events and $M$ is the number of reference $k$-mers. This can be improved using banded alignment, where the dynamic programming matrix is only filled in along the diagonal where the optimal alignment path is usually found. Tombo, Nanopolish and f5c both use adaptive banded alignment, where the band position is adjusted as alignment progresses to always be centered on the currently most probable path. Uncalled4 uses the standard DTW recursive cost function to fill the dynamic programming matrix with the addition of a multiplicative penalty for 'skips':

$$D[i,j] = \text{cost}(i,j) + \begin{cases} 0 & i = 0 \wedge j = 0 \\ \infty & i < 0 \vee j < 0 \\ \min(D[i-1,j-1], D[i-1,j], \text{skip} * D[i,j-1]) & \text{else} \end{cases}$$

where cost$(i,j)$ is the difference between the normalized mean current of event $i$ and the model $k$-mer at index $j$, and skip = 2 by default.

Uncalled4 uses a dynamic banding algorithm similar to that described by f5c (ref. 22), but the band placement is chosen before alignment begins using the basecaller 'moves' metadata. Basecallers such as Guppy and Dorado can optionally output 'moves' that represent approximate alignments between the signal and the basecalled read. These moves have low-resolution (five samples for DNA, ten for RNA), and often deviate from the true alignment by one or more reference positions. Uncalled4 projects these basecaller moves into reference coordinates based on the basecalled alignment cigar string, then centers the DTW bands on the ref-moves (Extended Data Fig. 2). This allows Uncalled4 to use a much narrower bandwidth (25 by default) than Nanopolish or Tombo, making alignment faster and preventing alignments from straying too far from the truth. Insertions and deletions (indels) that are larger than the bandwidth would cause a discontinuity in the band placement, so these are 'spliced' out of the read or reference, respectively, if they are above a threshold (ten by default) based on the ref-moves coordinates. Note Uncalled4 does not disrupt alignments over small indels, as these are a frequent basecaller error that can often be accurately aligned over. For deletions, the 'splicing' generates $k$-mers that are not present in the reference, which many downstream tools cannot handle, so these are masked and not included in the output by default. However, this can be disabled with the '–unmask-splice' option.

Uncalled4 encodes alignments as per-reference-coordinate statistics ('layers', below), at a minimum consisting of raw sample coordinates assigned to each site. This is unlike Nanopolish eventalign, which outputs multiple consecutive events aligning to the same reference position. The first step of most modification detection algorithms is to average these statistics on the nucleotide level, which is straightforward for the average current, but notably the current standard deviation cannot be accurately computed without re-analyzing the original raw signal. Uncalled4 outputs accurate per-nucleotide current standard deviations, which Tombo can also do via an optional flag. A consequence of this encoding is that 'stays' and 'skips' are not explicitly encoded by default. Skips can be identified by multiple consecutive reference positions having the same signal coordinates. Nanopolish masks skips and represents them as missing data. Uncalled4 penalizes

skips in the DTW cost function (2× for standard alignment, 4× for pore model training), and they can be masked via the '–mask-skips [all||keep_best]' option, which removes either all grouped positions or all but the closest to the reference current. Skips are masked during pore model training to reduce alignment errors, however, for modification detection we found that simply assigning the same values for each skipped position resulted in much higher recall with little change in precision. This reflects that modifications inherently disrupt signal alignment by deviating from the expected current, sometimes resulting in skips, and so masking skips removes useful information. This effect was most marked in RNA, where the motor speed is less consistent and skips may be caused by motor 'slippage'. Stay and skip rate can be computed by including the command 'uncalled4 align–count-events –tsv-out …', which includes the number of events aligned to each reference position. Counts greater than one indicate stays, while fractional counts indicate the inverse of the number of skipped positions (for example, 0.5 indicates two positions, 0.25 means two positions).

### Alignment encoding and formats

Uncalled4 represents each signal alignment as a set of per-reference-position statistics called alignment 'layers'. All alignments must include the 'length' layer, which indicates how many raw samples were aligned to each reference position, and usually include the current mean and standard deviation ('current' and 'current_sd'). Reference coordinates are defined relative to the 'central base' in the pore model, determined by the highest average per-position change in the model's substitution matrix (Fig. 2a). The current statistics are omitted for ref-moves due to their inaccuracy, and Tombo does not compute 'current_sd' by default. Additional layers can be derived from the base layers and/or the reference sequence, such as 'seq.kmer' (reference k-mer), 'dtw.model_diff' (absolute difference between the read current and model current) or 'mvcmp.dist' (distance from ref-moves) (Supplementary Table 6). Layer coordinates are defined relative to the pore model's central position determined by its substitution profile (Fig. 2a). Layers can also be offset from this position by a fixed number of reference coordinates (for example, dtw.dwell-11), facilitating generation of dwell time models at different pore offsets, for example (Fig. 2e). One minor limitation of this reference-oriented encoding is that Uncalled4 cannot output event-level statistics, but rather averages over multiple events aligned to the same position. Tombo stores alignments in a similar manner, while Nanopolish outputs per-event statistics. Most modification detection tools simply average these statistics over reference coordinates, and in doing so cannot accurately compute the true per-base current standard deviation without re-querying the raw signal file (that is FAST5, SLOW5 or POD5). Uncalled4 computes the true current standard deviation at each position, and can optionally output the number of events aligned to each position.

Uncalled4 primarily stores signal alignments in BAM tags, alongside the conventional basecalled alignments that were used to guide bcDTW. This format differs from Nanopolish basecalled read and alignment paths are fully preserved, and unlike f5c's similar format, Uncalled4 includes current means and standard deviations required for modification detection. This is accomplished in a space-efficient manner by storing the alignment layers in 16-bit integers. Raw signal coordinates ('us:' tag) are encoded as positive values indicating the number of aligned samples at each consecutive reference position, negative values indicating masked signal (that is no reference k-mers assigned to that stretch of signal), and zeros indicating 'skip' events (that is, no signal assigned to that reference position). Most positions fit within 16-bit integers, and for the few outliers that cannot, we reserve the maximum value of $2^{16}$-1 to be grouped with the subsequent length entry. Reference coordinates ('ur': tag) are encoded as a series of 'start' and 'stop' values indicating stretches of continuous alignment, with breaks caused by introns or deletions greater than '–del-max' (10 nt by default). The total span of the reference coordinate blocks should

be equal to the number of nonzero elements in the 'us:' tag, and the sum of the absolute values of 'us:' should equal the length of the raw signal. Current means ('uc:' tag) and standard deviations ('ud:' tag) are represented as 16-bit fixed-precision floating point values corresponding to normalized current levels ranging from −5.0 to 5.0 by default, representing a range of five standard deviations from the mean. Masked positions can be assigned a 'null' value equal to $-2^{16}$-1, allowing for representation of masked signal aligned to a mask reference position, which is necessary to mask large insertions or deletions and to represent pervasive masking in Nanopolish alignments. Normalized units can be converted to picoamps, or whichever units are defined by the pore model, using parameters stored in JSON format in the BAM header. This JSON header stores additional information on the tag labels, fixed-point scaling factors, reference and raw signal paths, and other pore model metadata. Normalization parameters are stored in the 'un' tag (that is scale and shift), which can be used to linearly scale the calibrated raw signal into the normalized signal that corresponds to the normalized mean values in the 'uc' tag. The addition of the signal alignment tags increases the BAM file size by 2–4-fold, but is still several times smaller than the Nanopolish and Tombo formats (Table 1).

In addition to the BAM format, Uncalled4 supports two text-based output formats: 'eventalign' and 'TSV'. Eventalign is based on Nanopolish's default tab-delimited output format, and is mainly included for compatibility with modification detection tools such as xPore and m6Anet. 'TSV' is a customizable tab-delimited format, which can include any of the alignment layers or comparison statistics (below). These formats can be written directly by the 'uncalled4 align' command, or can be derived from an Uncalled4 BAM file via the 'uncalled4 convert' command with the '–bam-in' option. 'uncalled4 convert' can also convert Nanopolish eventalign files or Tombo FAST5 files into the BAM format, which is necessary for analysis and visualization of these alignments by Uncalled4.

Finally, to expedite m6Anet analysis and demonstrate the utility of the BAM alignment format, Uncalled4 includes a conversion function from a sorted BAM file to the m6Anet 'dataprep' format that collects signal features in a per-reference-coordinate JSON format. This can also be accomplished by first converting the BAM file to 'eventalign' format and using 'm6anet dataprep', however we found conversion from eventalign format was by far the largest bottleneck in m6Anet analysis. The sorted BAM format enables conversion in a single linear read of the file, making conversion many times faster than the random parallel file access required to convert from eventalign, especially on a shared compute cluster where parallel disk access is slow.

### Analysis of signal alignments

Uncalled4 can perform analysis on any signal alignments in the BAM format, which can be divided into reference-level (refstats command), read-level (readstats command) and read-base-level (convert and compare). Reference-level analysis includes simple summary statistics such as mean and standard deviation of current levels and dwell times, or comparative statistics such as KS statistics between two samples. Similarly, read-level analysis outputs summary statistics of layers over entire reads, or segments of reads defined by reference coordinates. Basic and derived layers of individual reads at each reference coordinate can be output in TSV format via the convert command. If basecaller moves are included in the BAM file, this can include ref-moves distance metrics (described below).

Uncalled4 can compare two different alignment methods applied to the same set of reads by inputting two sorted signal alignment BAM files to the compare command, producing a table of per-reference-coordinate signal Jaccard distances and signal-to-reference distances. These can also be visualized via the dotplot command (Fig. 1c). Signal Jaccard distance, the inverse of the Jaccard similarity, measures the degree of overlap between raw samples aligned to each reference coordinate: $1 − (A \cup B)/(A \cap B)$, where A and B

are the sets of raw samples aligned to each reference coordinate. This varies between 0 and 1, where 1 indicates no overlap and 0 indicates perfect overlap. Signal-to-reference distance measures the average number of reference bases between the raw samples aligned to each coordinate. This is computed reciprocally for each method, computing the nucleotide distance for each raw sample aligned to each reference coordinate and then averaged between the two methods. These metrics can also be computed between a signal alignment method and the ref-moves used to guide Uncalled4 alignment, via the convert command or in any of the visualizations. In this case, signal-to-reference distance is not computed reciprocally, instead only averaging over the alignment method raw samples and not the low-resolution ref-moves.

## Visualizations

All visualizations produced directly by Uncalled4 are implemented in Python using Plotly, which produces interactive web browser-based plots that were exported to SVG format. The three main alignment visualizations (trackplot, dotplot and refplot) and also integrated into an interactive signal genome browser using Dash, a local web server designed for Plotly. Pore model profiles (Fig. 1a) are generated using pore models only by computing the absolute change in current generated substituting each base for each other base at every possible $k$-mer. This is efficiently implemented in Python and C++ using the 'buffer protocol', which allows for vectorization of $k$-mer operations. Some figures were also generated by the Python matplotlib library, in cases where reproducible interactivity is not necessary.

IGV visualizations were generated by encoding the per-read modifications defined m6Anet's 'data.indiv_proba.csv' file into BAM modification tags. The reference coordinates were translated to read coordinates using the cigar string, and positions where an 'A' was not present in the read sequence were excluded. Site-level probabilities were multiplied by a constant factor (4) for visualization purposes. The IGV screenshots were exported to the SVG format and edited in Inkscape for clarity.

## Pore model training

Pore model training is an iterative process, where in each iteration reads are aligned until every $k$-mer is represented a minimum number of times (500 by default), after which summary statistics (median and standard deviation) of signal characteristics (current mean, current standard deviation and dwell time) are recorded and used as the pore model for the next iteration. In each training iteration, only positions with low signal-to-reference distance to ref-moves are included (mvcmp.dist <= 1, by default), which eliminates many alignment errors. Only one normalization iteration is applied during training, since linear regression is sensitive to outliers that are frequent in early training iterations. A higher skip penalty is also used for pore model training, and skipped positions are masked to further reduce alignment errors (equivalent to 'uncalled4 align –skip-cost 4 –mask-skips keep_best'). This process requires an initial 'draft' model to use in the first iteration. This draft model could be a canonical nucleotide model, for example with the goal of retraining it for modified nucleotides. We also developed a de novo initialization method, not requiring a previous $k$-mer pore model beforehand, using the ref-moves used in Uncalled4's bcDTW algorithm. The ref-moves can be treated as standard signal alignments, although they are frequently one or two bases from their true position. To mitigate these inaccuracies, we began training using a short $k$-mer length to average-out the initial errors (1-mers for r9.4, 4-mers for r10.4) and increased the $k$-mer length every $n$ training iterations (two iterations for DNA, three for RNA) until the desired $k$-mer length was reached.

The Uncalled4 train subcommand runs the training procedure for a specified $k$-mer length and number of iterations. It outputs a directory with the pore model for each iteration in a binary NumPy format, along with indexed alignment statistics used to generate each model. To progressively increase the $k$-mer length, the command was run once per-$k$-mer length, each time using the last pore model output in the previous iteration as the new initialization model. This training procedure is flexible, allowing for alternate initialization methods or $k$-mer expanding methods to be tested. We evaluated the effectiveness of each training procedure based on the Pearson's correlation coefficient between the trained model and ONT's pore models. Intermediate models with shorter $k$-mer lengths can also be evaluated by 'reducing' ONT's models by averaging the values of $k$-mers that share central bases, implemented in Uncalled4's 'PoreModel.reduce' method.

The align subcommand will attempt to automatically detect the appropriate pore model to use based on metadata in the raw signal FAST5/POD5/SLOW5 files. If this cannot be detected, the user can specify a preset pore model ('dna_r10.4.1_400bps_9mer', 'dna_r9.4.1_400bps_6mer' or 'rna_r9.4.1_70bps_5mer') using '–pore-model' flag or by defining the '–flowcell' and '–kit' used for basecalling or a custom pore model can be provided. The default r9.4.1 DNA and RNA pore models are provided by ONT (https://github.com/nanoporetech/kmer_models) under 'legacy' models, while for r10.4.1 DNA we use the Uncalled4 model trained on unmodified *D. melanogaster* data, as described above.

## Modification detection, training and assessment

Comparative KS statistics were computed by the Uncalled4 refstats command ('uncalled4 refstats current.mean ks …'), which uses the Python Scipy (v.1.10.1) package's 'ks_2samp' function to compute two-sample KS statistics over per-$k$-mer mean currents between two samples. For Tombo alignments, we compared the output to the Tombo KS statistic output, with highly similar results. xPore was run on Nanopolish using recommended parameters, and on Uncalled4 via conversion to the eventalign format. Transcript-level modification calls were translated to the gene level using a custom Python script ('t2g.py') provided with Uncalled4 (Extended Data Fig. 8), which uses a GTF annotation to add gene IDs and coordinates to a tab- or comma-delimited file, and these values were averaged over each gene using the Pandas 'groupby' operation to take the mean of site-level probabilities. Unless otherwise specified, only sites with at least 20× coverage by basecalled reads were considered for modification detection.

m6Anet was trained for Uncalled4 alignments on the HCT116 cell line from the Singapore Nanopore Expression Project (replicate 3)[48], which was originally used for the default m6Anet model[31]. These data were re-basecalled using Guppy v.6.4.8, and we also retrained Nanopolish to assess the effect of re-basecalling. m6A labels obtained from ref. 31 were provided in transcript coordinates, with sites divided by-gene into 'train', 'test' and 'validation' sites. The recommended training procedure only included primary transcriptome alignments, and we noted that many reads aligned to a different transcript from the same gene than was listed in the training data. We therefore mapped the training data to gene-level coordinates, then back to transcript level using the transcripts present in the re-basecalled data, maintaining the same 'train', 'test' and 'validation' gene assignments.

Precision-recall curves and receiver operator characteristic (ROC) curves were visualized and the corresponding area under the curve were computed using Scipy. Both these metrics measure recall, also known as true positive rate, defined as TP/(TP + FN) (TP is true positive count, FN is false-negative count). Accurate estimation of false negative is complicated by prefiltering performed by tools such as xPore and m6Anet, where many sites are not assigned a probability and excluded from the output. We found many of these sites are actually modified, and so not counting these decreases the false-negative count and thus falsely increases the recall (Fig. 4b and Extended Data Fig. 7). We used two strategies to compensate for this. For the transcript-level HEK293T results, we computed coverage from minimap2 alignments of basecalled reads, only considering read endpoints and not internal deletions, and included all sufficiently covered (20×) sites by filling in

a probability of zero. For the gene-level results, different alignment strategies cover different sites, and so we simply took the union of all sites covered by each tool and filled missing values with probability zero. Many sites with probability zero can generate a large 'jump' at the end of precision-recall and ROC curves, making the area under the curve estimate less informative. For precision-recall, we use the 'average precision' definition of AUPRC, which essentially treats the curve as a stepwise function rather than using the trapezoidal rule for area calculation, which is more robust to skewed datasets. There is no comparable alternative for AUROC, but visual inspection suggests the overall trends would be the same regardless.

## Data processing

All nanopore read data were basecalled with Guppy v.6.4.8 using high-accuracy models with the '−moves_out' option, with the exception of Tombo alignments where an earlier version of Guppy (v.6.0.1) that supported output of basecalled FAST5 files required for Tombo. 5mCpG calling was also performed using the Guppy 5mCpG high-accuracy model. Reads were aligned using Guppy's builtin minimap2 alignment option, which encodes the 'moves' basecaller metadata in primary alignment tags. We also provide a Python script that copies these tags into supplemental and secondary alignments for efficient alignment with Uncalled4 ('bamprep.py'). To re-align reads while preserving basecaller metadata, for example to compare spliced genome alignments with transcriptome alignment, we converted the primary BAM alignments to FASTQ with the relevant tags in the header using the command 'samtools fastq -T mv,ts', then aligned using minimap2 (v.2.16) with the '-y' option to copy tags from the headers. Coverage of minimap2 basecalled read alignments was computed using the bedtools[49] command 'bedtools genomecov -d -pc' to count the number of reads overlapping each adenine or DRACH site, only considering read start and end sites and not internal deletions.

The *D. melanogaster* data were aligned to the *D. melanogaster* ISO1 release 6 reference genome (RefSeq GCF_000001215.4). The *E. coli* rRNA data were aligned to the 16S and 23S transcripts from the *E. coli* transcriptome (GenBank NC_000913.3), with modification labels obtained from ref. 28. All human datasets were aligned to a transcriptome derived from GRCh38 Ensembl annotations v.91 obtained from ref. 30, or directly aligned to GRCh38 for spliced genome alignments (GCF_000001405.26). HEK293T METTL3-sensitive m6A labels were also obtained from ref. 30. HCT116 m6A training labels were obtained from ref. 31. All other m6A labels are from the m6A-Atlas v.2 (ref. 15) (accessed 12 May 2023).

Uncalled4 signal alignments were compared to Nanopolish (v.0.13.3), Tombo (v.1.5.1) and f5c (v.1.3). Nanopolish and f5c were run using the '−scale-events−signal-index' options, which are required for Uncalled4, xPore and m6Anet. Timing was measured using a single central processing unit thread, and f5c additionally used a Nvidia Quadro P5000 GPU with default parameters. KS statistics were primarily computed with Uncalled4, and produced similar results to Tombo's builtin KS statistic output, with minor differences attributed to read filtering and rounding error. We also compared Uncalled4 and Nanopolish RNA modification detection performance using xPore (v.2.1) and m6Anet (v.2.0.2). m6Anet was retrained using Uncalled4 and Nanopolish alignments on re-basecalled HCT116 cell line data and labels originally used for m6Anet. GNU parallel was also used to efficiently run tasks in parallel.

## DNA extraction

Genomic DNA was extracted from 15 newly eclosed *D. melanogaster* males of the Oregon-R strain (Bloomington stock center number 5, RRID BDSC_5). After selection, males were immobilized by freezing at −80 °C for 5 min. Next, the flies were crushed with a pipette tip in 200 μl of Buffer A (100 mM Tris-HCl, pH 7.5, 100 mM EDTA, 100 mM NaCl, 0.5% SDS). This was followed by a 30-min incubation at 65 °C. After incubation, 400 μl of KOAc:LiCl (prepared by combining one part of 5 M KOAc

with two parts of 6 M LiCl) was added and the mixture was allowed to precipitate on ice for 10 min and then the precipitate was pelleted at room temperature at 14,000 rpm for 15 min. The supernatant containing nucleic acids was transferred to a clean microcentrifuge tube and isopropanol was added at a ratio of 600 μl per 1 ml of supernatant. The DNA was then precipitated by centrifuging at 14,000 rpm for 15 min at room temperature. Afterward, the supernatant was removed, and the pellet was washed with 1 ml of cold ethanol (70–75%). The pellet was then centrifuged again for 5 min before removing the ethanol wash. After air drying, the pellet was resuspended in ultra-pure water.

## Genomic DNA shearing and amplification

*D. melanogaster* genomic DNA (roughly 500 ng) was diluted into a total volume of 49 μl of ultra-pure water. To shear the genomic DNA to 8-kb fragments, DNA was transferred to a g-Tube (Covaris, 520079) and centrifuged at room temperature for 1 min at 6,000 rpm. The g-Tube was then inverted and centrifuged again at room temperature for 1 min at 6,000 rpm. Centrifugation was carried out on an Eppendorf Centrifuge 5425 (Eppendorf, 5405000646).

Sheared DNA was then amplified using the ONT protocol for low-input PCR (low-input-genomic-dna-with-pcr-sqk-lsk110-LWP_9117_v110_revJ_10Nov2020-minion). First, the sheared DNA was mixed with NEBNext Ultra II End Prep Enzyme Mix and Reaction Buffer (NEB, E7180S) and incubated at 20 °C for 5 min followed by 65 °C for 5 min to repair fragment ends. DNA was then purified using 1× AMPure XP beads (Beckman Coulter, A63881) along with 70% ethanol. DNA was eluted in 31 μl of nuclease-free water and quantified with the Qubit broad range double-stranded DNA (dsDNA) assay (ThermoFisher Scientific, Q32850). Next, end-prepped fragments were mixed with PCR adapters (ONT, EXP-PCA001) and Blunt/TA Ligase Master Mix (NEB, M0367S) and incubated for 15 min at room temperature. DNA was then purified using 0.4× AMPure XP beads and 70% ethanol. DNA was eluted in 26 μl of nuclease-free water at room temperature and quantified with the Qubit broad range dsDNA assay. DNA was then diluted to 10 ng μl⁻¹ in water.

Twelve PCRs were then performed by combining 20 ng of diluted DNA, 46 μl of water, 2 μl of Primer Mix (ONT, EXP-PCA001) and 50 μl of LongAmp Taq 2× master mix (NEB, M0287S) (Supplementary Table 7).

Pairs of PCR reactions were then combined and DNA was purified using 0.4× AMPure XP beads and 70% ethanol. DNA was eluted at room temperature in 30 μl of nuclease-free water and quantified with the Qubit broad range dsDNA assay. Fragment size was quantified using a Genomic DNA ScreenTape (Agilent, 5067–5365) on a TapeStation 4200 (Agilent, G2991BA). All DNA was then pooled and stored at −20 °C until use.

## 5mCpG labeling

Labeling of CpGs with 5mC to create a training dataset was performed similarly to previous work[21,50]. Two labeling reactions were set up as follows. Amplified DNA (4 μg) was combined with 40 μl of water, 8 μl of 10× NEB Buffer 2 (NEB, B7002S), 8 μl of 1.6 mM S-adenosylmethionine (SAM) (NEB, B9003S) and 16 units of M.SssI (NEB, M0226S). Reactions were incubated for 4 h at 37 °C. After 2 h of incubation, both 1.6 mM SAM (8 μl) and M.SssI (16 units) were added to the reactions to replenish enzyme activity. DNA was then purified using 0.8× AMPure XP beads along with 70% ethanol and eluted in 22 μl of nuclease-free water. DNA was then quantified with the Qubit broad range dsDNA assay and fragment size was quantified using a Genomic DNA ScreenTape on a TapeStation 4200. DNA from the two reactions were then pooled and stored at −20 °C until further use. A second round of M.SssI labeling on the pooled DNA was performed identically to the labeling reaction above. After final DNA purification and quantification, the DNA was stored at −20 °C until sequencing.

## Nanopore library preparation

Four ONT sequencing runs were performed. Both unlabeled and labeled DNA were sequencing on r9.4.1 pores as well as r10.4.1 pores at 400bps.

Libraries for r9.4.1 pores were constructed using the LSK110 ligation sequencing kit (ONT, SQK-LSK110) and r10.4.1 pore libraries were constructed using the LSK114 ligation sequencing kit (ONT, SQK-LSK114). Both LSK110 (genomic-dna-by-ligation-sqk-lsk110-GDE_9108_v110_revV_10Nov2020-minion) and LSK114 (genomic-dna-by-ligation-sqk-lsk114-GDE_9161_v114_revG_29Jun2022-minion) have similar protocols so only one set of steps will be described below with notes on kit specific changes.

First, DNA fragments (1.25 µg) were mixed with NEBNext Ultra II End Prep Enzyme Mix and Reaction Buffer and incubated at 20 °C for 5 min followed by 65 °C for 5 min to repair fragment ends. DNA was then purified using 1× AMPure XP beads along with 70% ethanol, eluted in 61 µl of nuclease-free water at room temperature, and quantified with the Qubit broad range dsDNA assay. End-prepped DNA was then mixed with Ligation Buffer (ONT, SQK-LSK110 and SQK-LSK114), NEBNext Quick T4 DNA ligase (NEB, E7180S), and either Adapter Mix F for r9.4.1 pores (ONT, SQK-LSK110) or Ligation Adapter for r10.4.1 pores (ONT, SQK-LSK114). Reactions were incubated for 15 min at room temperature. DNA was purified using 0.4× AMPure XP beads and Long Fragment Buffer (ONT, SQK-LSK110 and SQK-LSK114). DNA was eluted in 15 µl of Elution Buffer (ONT, SQK-LSK110 and SQK-LSK114) at 37 °C for 10 min. DNA was then quantified with the Qubit broad range dsDNA assay.

### Nanopore sequencing
R9.4.1 pore sequencing was performed using ~40–50 fmol of library on r9.4.1 MinION flow cells (ONT, FLO-MIN106D). R10.4.1 pore sequencing was performed using ~20 fmol of library on r10.4.1 MinION flow cells (ONT, FLO-MIN114) using either 260 or 400 bases per second mode. According to the manufacturer's recommendations, bovine serum albumin (Invitrogen, AM2616) was added to the sequencing flush buffer at a concentration of 0.2 mg ml$^{-1}$ for all r10.4.1 flow cell sequencing runs. All sequencing runs were performed on a GridION Mk1 sequencing device (ONT, GRD-MK1) and run for 72 h.

### Reporting summary
Further information on research design is available in the Nature Portfolio Reporting Summary linked to this article.

### Data availability
The *D. melanogaster* ONT sequencing data described above are deposited on the sequence read archive (SRA) bioproject PRJNA1082764. All other datasets were obtained from publicly available sources. The *E. coli* rRNA data were obtained from ref. 28 (SRA bioproject PRJNA634693). The constitutively incorporated BrdU dataset and matched control were from ref. 29 (SRR8991355 and SRR8991351). The IVT HeLa cell direct RNA sequencing data used to train the RNA002 model were obtained from ref. 50 (SRR23950400). HEK293T wild-type and METTL3 knockouts were obtained from ref. 30 (PRJEB40872). NA12878 data were obtained from ref. 51 (https://github.com/nanopore-wgs-consortium/NA12878). HMEC data were obtained from ref. 52 (GEO accession GSE132971). All other human cell line data are from the Singapore Nanopore Expression Project (PRJEB40872, Supplementary Table 2).

### Code availability
Uncalled4 is available open source at github.com/skovaka/uncalled4. Trained pore models, the Uncalled4 m6Anet model and data used to generate the main figures are available at https://github.com/skovaka/uncalled4_supplemental_data.

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

## Acknowledgements
We thank J. A. Urban for the extraction of *D. melanogaster* genomic DNA and helpful discussion. We thank B. Langmead, O. Ahmed, M. Zakeri, M. Pertea, H. Gamaarachchi, A. Wassie and J. Yang for their helpful discussions. This work was supported, in part, by National Institutes of Health grant award numbers U01CA253481, U01HG013744 and OT2OD002751 (to M.C.S.), NSF grant award number IOS-2216612 (to M.C.S.), grant numbers NHGRI HG010538 and HG009190 (to W.T.), The Lustgarten Foundation grant award number 90101412 (to M.C.S.) and the Commonwealth Foundation (to M.C.S.). Part of this research project was conducted using computational resources at the Maryland Advanced Research Computing Center (MARCC).

## Author contributions
S.K. and M.C.S. designed Uncalled4. S.K. implemented Uncalled4. S.K. and V.S. benchmarked Uncalled4. P.W.H. performed the *D. melanogaster* library preparation and sequencing. P.W.H. and L.B.M. performed alignment and other genomic analyses for the *D. melanogaster* data. S.K. ran all signal alignments, modification detectors and statistical analyses. K.M.J. advised on the analysis and biological interpretation of the data. W.T., P.W.H., L.B.M. and R.R. advised on applications for nanopore signal analysis. W.T. supervised sequencing runs and advised on the experimental designs. M.C.S. supervised the entire project. All authors contributed to writing and editing the paper. All authors read and approved the final paper.

## Competing interests
W.T. has two patents (8,748,091 and 8,394,584) licensed to ONT. S.K. has received travel funding from ONT. The other authors declare no competing interests.

## Additional information
**Extended data** is available for this paper at https://doi.org/10.1038/s41592-025-02631-4.

**Correspondence and requests for materials** should be addressed to Sam Kovaka.

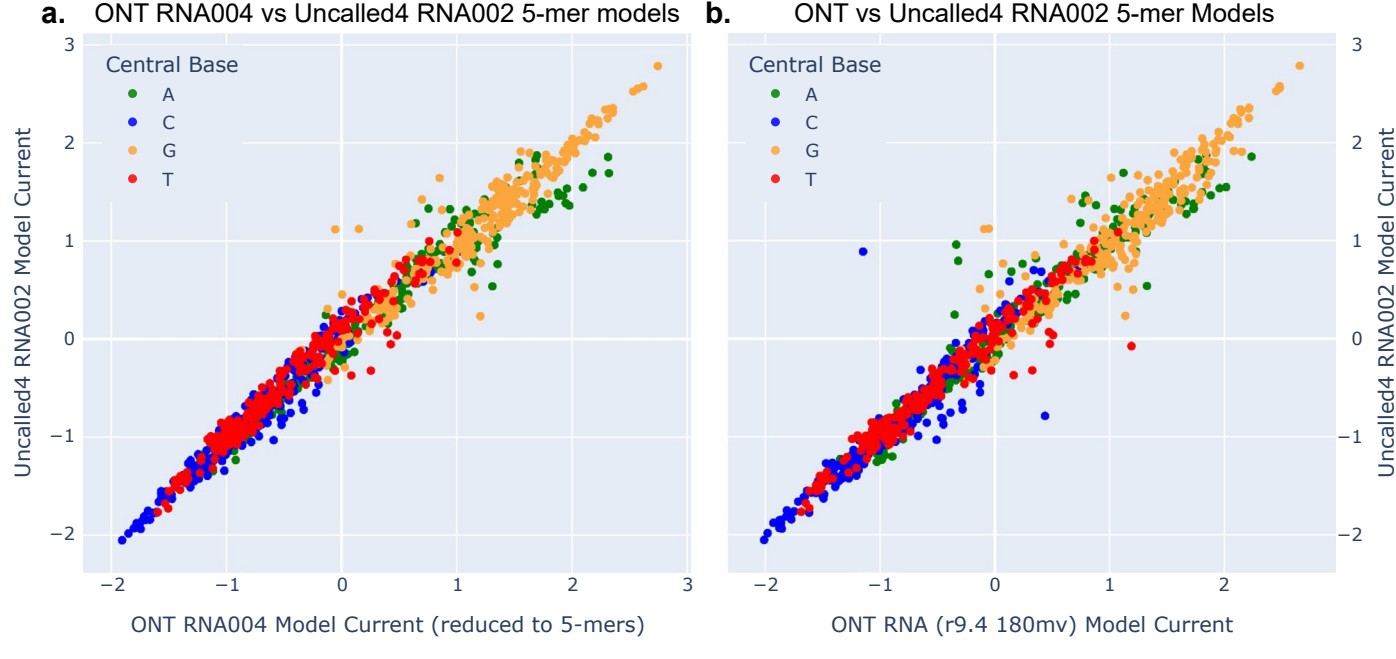

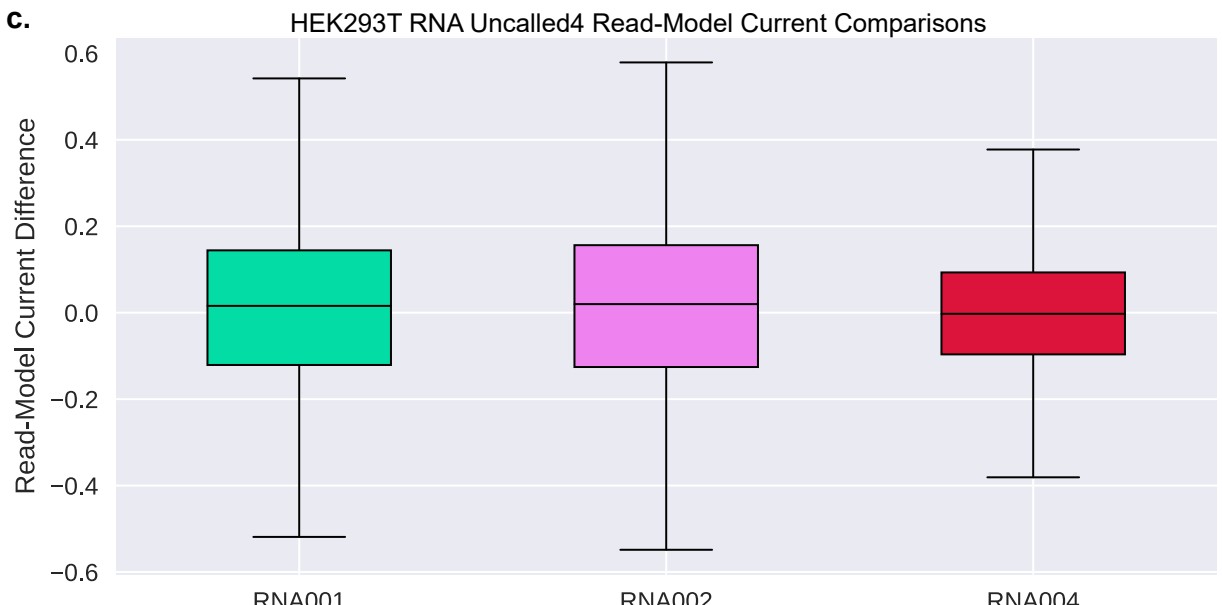

**Extended Data Fig. 1 | Comparisons between RNA pore model per-k-mer current means. (a)** Comparison between the five central bases of ONT's 9-mer RNA004 model and an Uncalled4-trained RNA002 5-mer model. **(b)** Uncalled4-trained RNA002 model compared with and ONT 'rna_r9.4_180mv_70bps' model, which is the default model that Uncalled4 and Nanopolish use for RNA001 or RNA002. **(c)** Boxplots showing distribution of differences between the mean current of signal aligned to the HEK293T reference and the current predicted by the k-mer model. Boxes span the first and third quartiles with the median indicated by the horizontal line, and whiskers extend to 1.5 times the interquartile range.

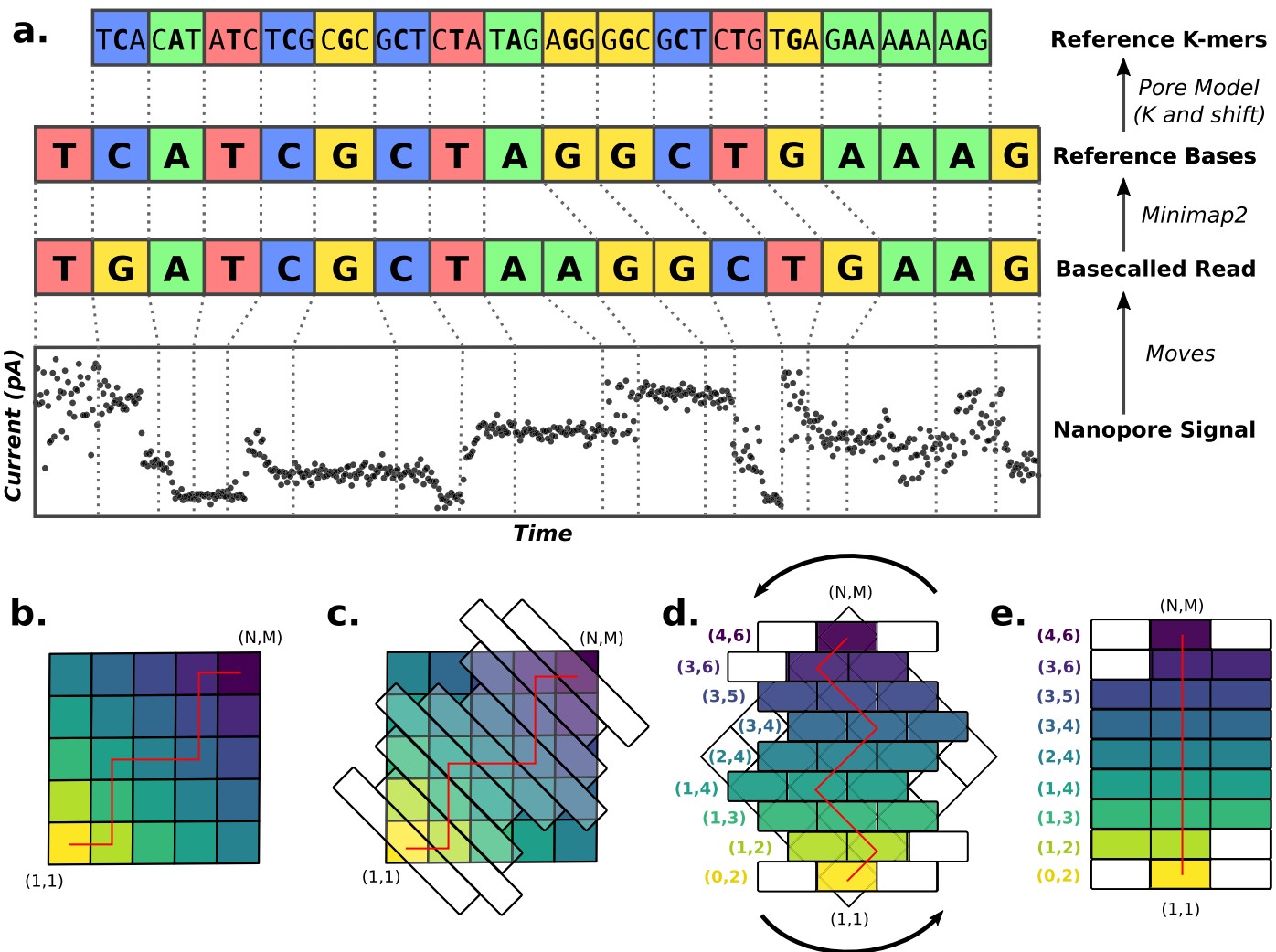

**Extended Data Fig. 2 | Illustration of basecaller-guided DTW. (a)** Generating of *ref-moves* from raw basecaller *moves* and a minimap2 alignment. The minmap2 'CIGAR' corresponding to the basecalled read alignment is '9M1I6M1D3M'. K-mers coordinates are defined relative to the central base, which is defined for each pore model based on its substitution matrix (Fig. 2a). **(b)** A standard NxM DTW matrix, where N = M = 5. Cells are colored by their Manhattan distance from (1,1), which corresponds to the band which they will be contained in. The red line represents the *ref-moves* which will guide band placement. **(c)** The same DTW matrix overlaid with bands centered on the *ref-moves* (band width W = 3). **(d)** The DTW band matrix with each row offset by its location in the NxM matrix, which is shaded in the background and rotated 45o. White cells indicate out-of-bounds coordinates. Band start coordinates are indicated by the colored numbers to the left. **(e)** The DTW band matrix, represented as a standard two-dimensional array. Note that the start coordinates are required to reconstruct the original matrix structure.

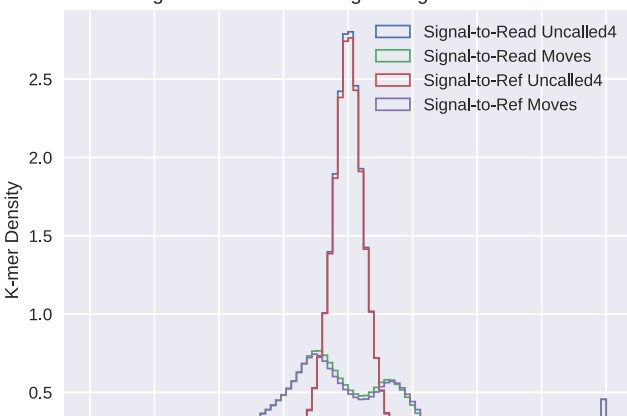

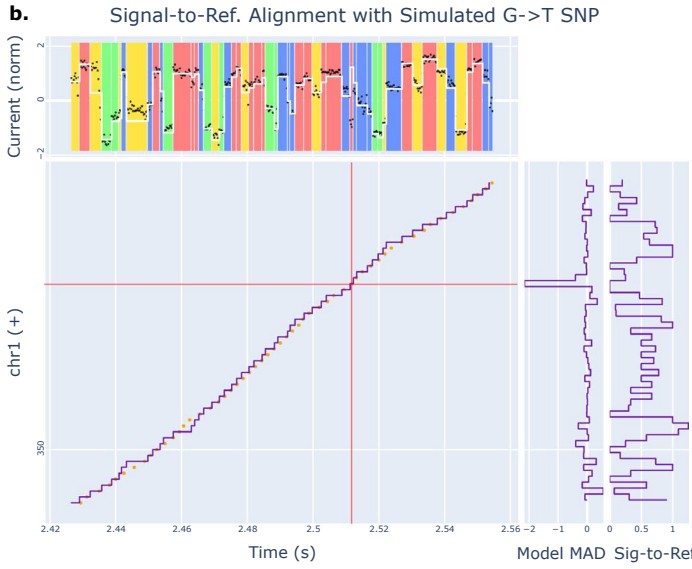

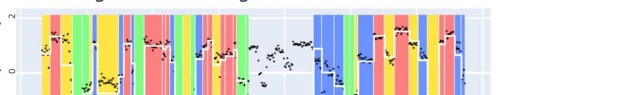

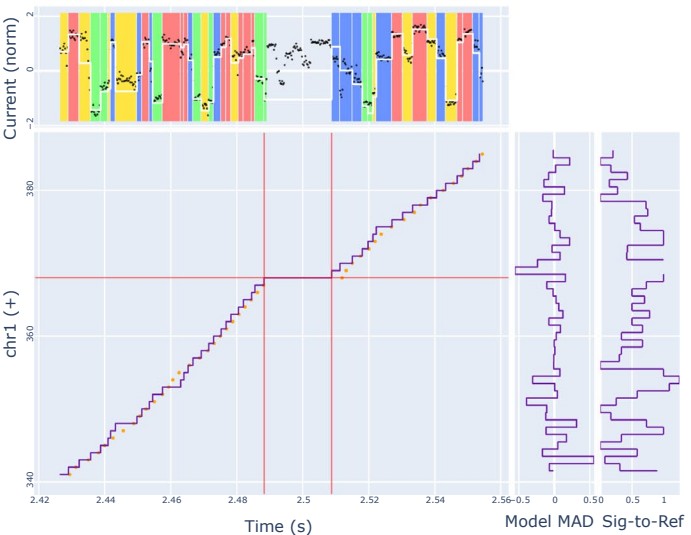

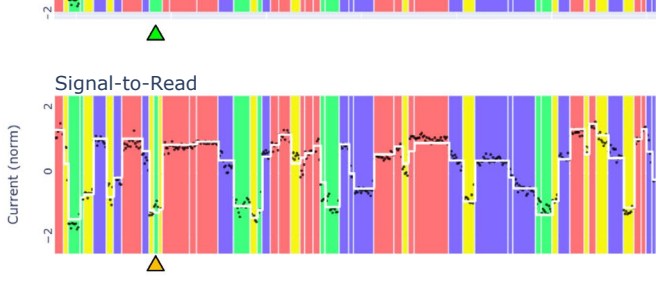

**Extended Data Fig. 3 | Signal-to-read and signal-to-reference alignment.**
**(a)** Per-k-mer current means from signal-to-read and signal-to-reference
Uncalled4 alignments and uncorrected basecaller moves. **(b)** Alignment dotplot
of a *D. melanogaster* r10.4.1 read to a reference containing a spiked-in G- > T
substitution at the location indicated by the red lines, causing increased read-
model current MAD. **(c)** Alignment dotplot of the same read to a reference with a
10 nucleotide deletion with boundaries indicated by red lines. Uncalled4 masks
signal around insertions or deletions 10 nucleotides or larger based on the *ref-*
*moves* coordinates, meaning the signal corresponding to the deleted sequence
is not included. **(d)** Reference- and read-aligned signal of a read which features
a likely sequencing error causing a two nucleotide insertion in the basecalled
sequence. A slight jump in signal is observed within the signal mapping to 'A' in
the signal-to-reference alignment, indicated by the green arrow. This nucleotide
is broken into 'GAG' in the basecalled sequence, making the signal-to-read
alignment erroneously more similar to the pore model.

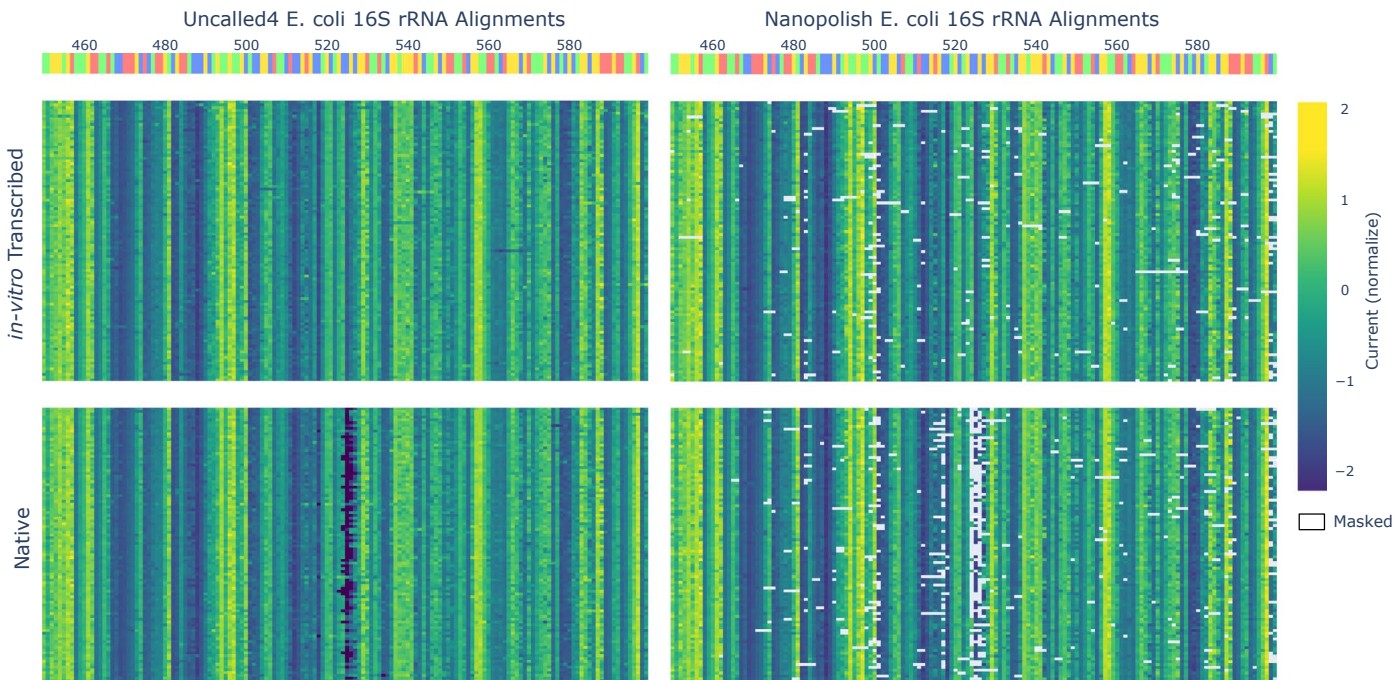

**Extended Data Fig. 4 | Modification signal trackplots.** Trackplots displaying per-read-k-mer mean current levels for Uncalled4 and Nanopolish *in-vitro* transcribed and native *E. coli* ribosomal RNA sequenced with ONT RNA002. An O6-methylguanine site is present in the native dataset at position 526, causing a drop in current. White cells indicate masked positions, where Uncalled4 performs no masking in this dataset because there were no large insertions or deletions, while Nanopolish masks many positions particularly around the modification site.

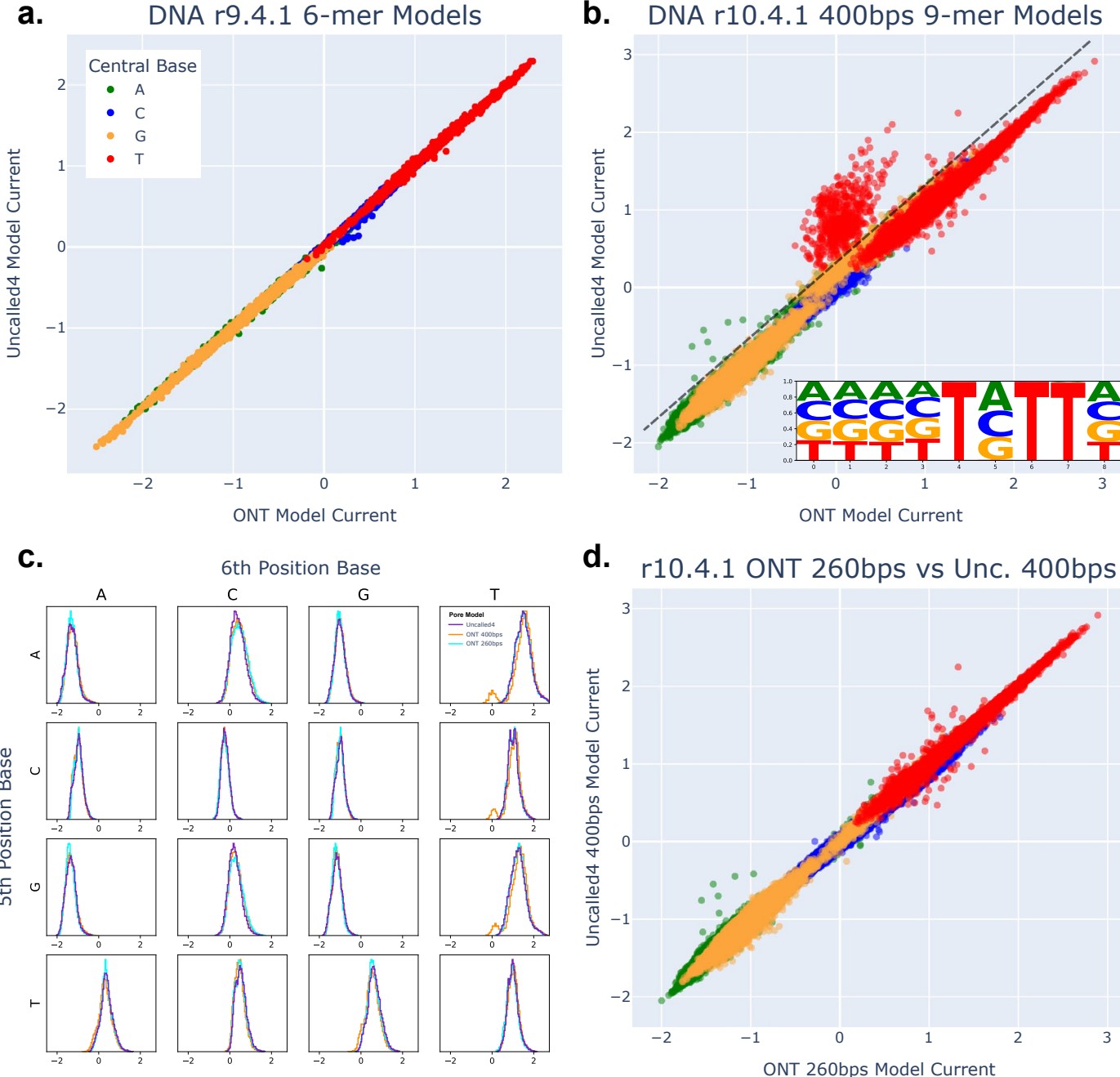

**Extended Data Fig. 5 | DNA model training results. (a)** Current levels from Uncalled4 and ONT r9.4.1 6-mer DNA models. **(b)** Current levels from Uncalled4 and ONT r10.4.1 400 bps 9-mer DNA models. Inset displays sequence logo for k-mers with more than 0.5 normalized units of difference between the models (indicated on main plot by dashed line). **(c)** Current distributions for k-mers with each base fixed at the 6th and 5-th positions for r10.4.1 models, including both 400 bps and 260 bps ONT models. Most distributions are unimodal, except for ONT 400 bps which has outliers caused by 'TVTT' k-mers. **(d)** Comparison between Uncalled4's r10.4.1 400 bps model and ONT's 260 bps model, which lacks the outliers seen in ONT's 400 bps model.

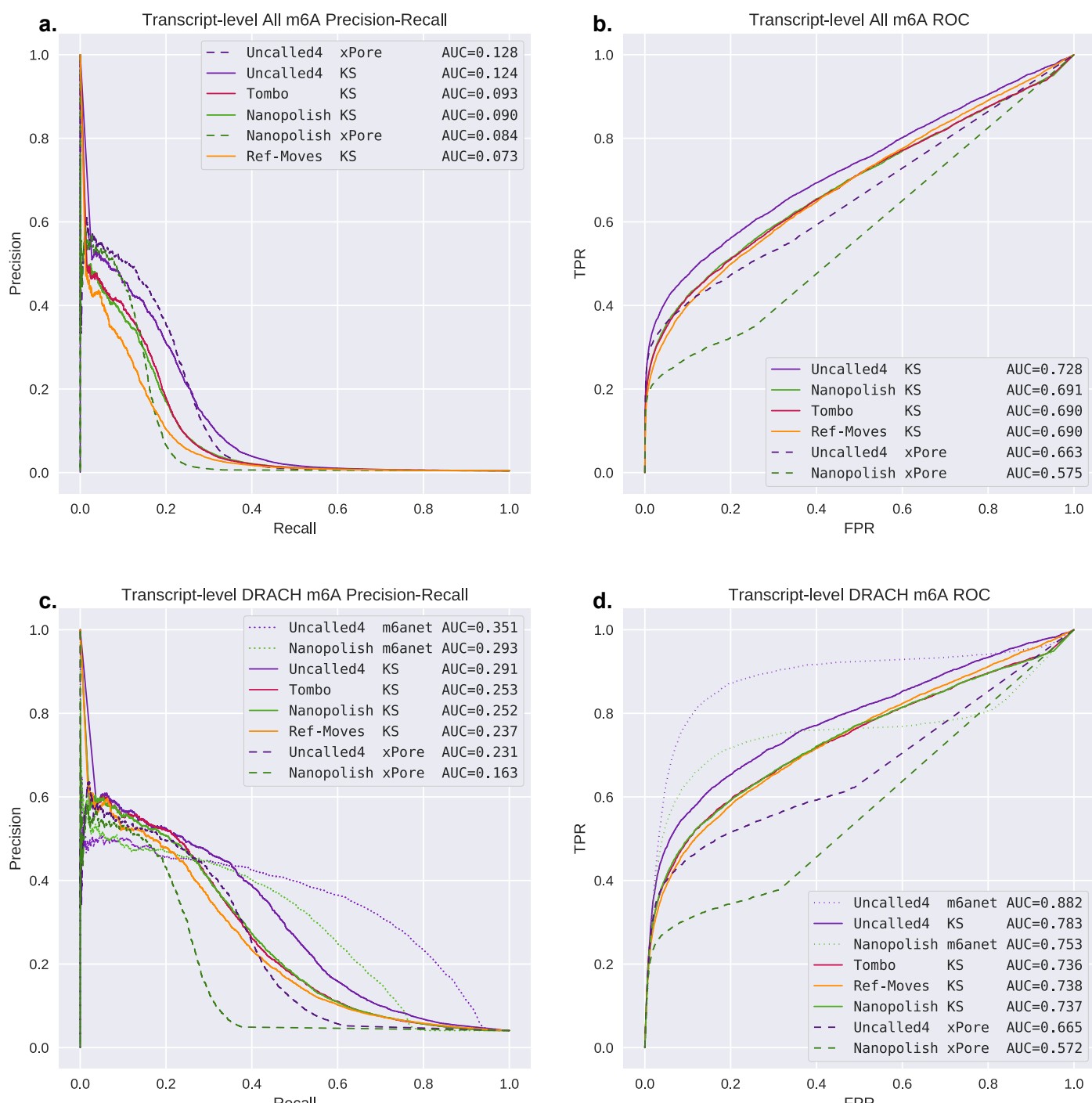

**Extended Data Fig. 6 | Transcript-level comparative m6A detection.** Precision recall and ROC curves for transcript-level comparative m6A detection in HEK293t in all contexts (**a-b**) and limited to DRACH sites (**c-d**).

**a.**

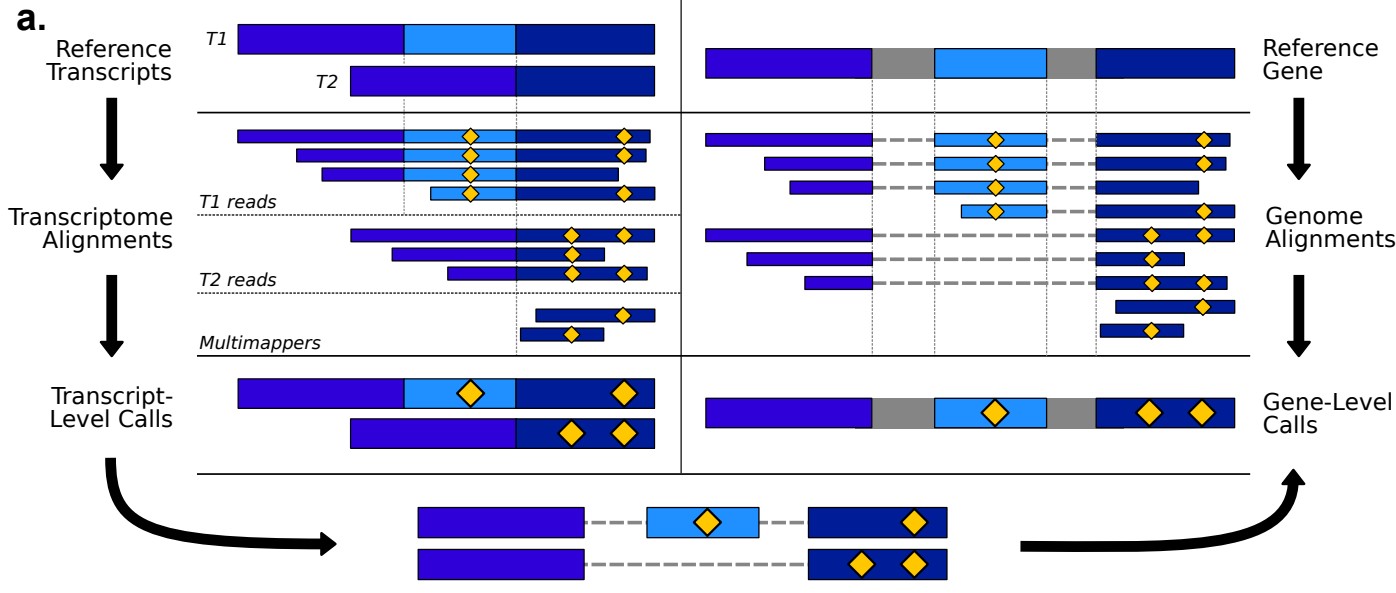

**b.**

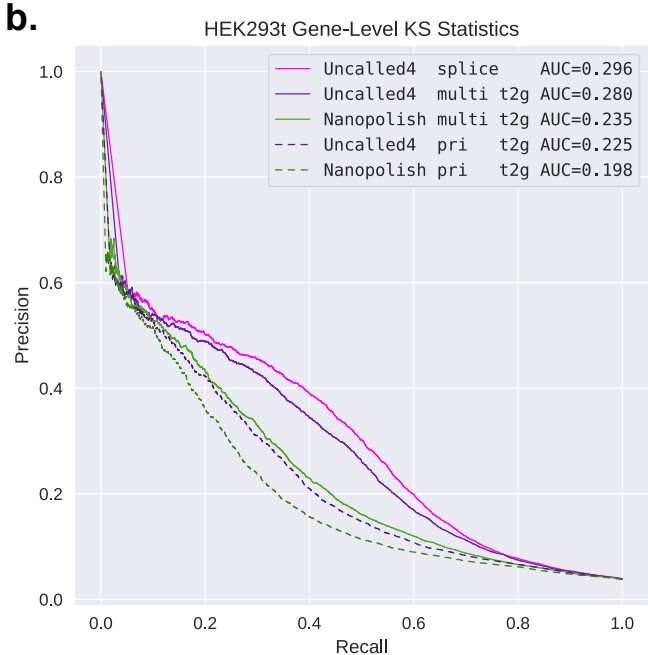

**c.**

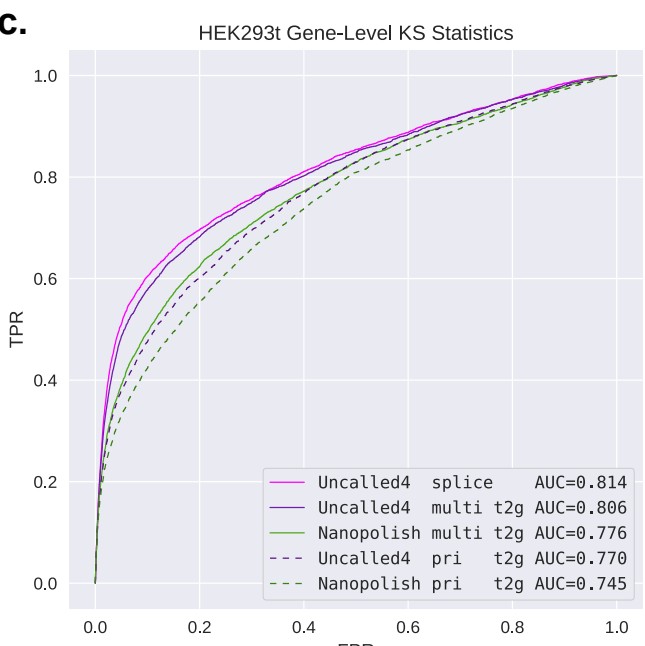

**Extended Data Fig. 7 | Gene-level modification detection methods.**
**(a)** Illustration of transcript-level modification calling, genome-level calling, and translation of transcript-level calls to the gene-level (t2g). **(b)** Precision-recall and **(c)** ROC curves of Uncalled4 and Nanopolish gene-level calls using KS statistics.

'Splice' indicates Uncalled4 spliced genome alignment. 'Multi t2g' indicates transcript-to-gene averaging using all multi-mapping reads, while 'pri t2g' indicates the same but only using primary alignments.

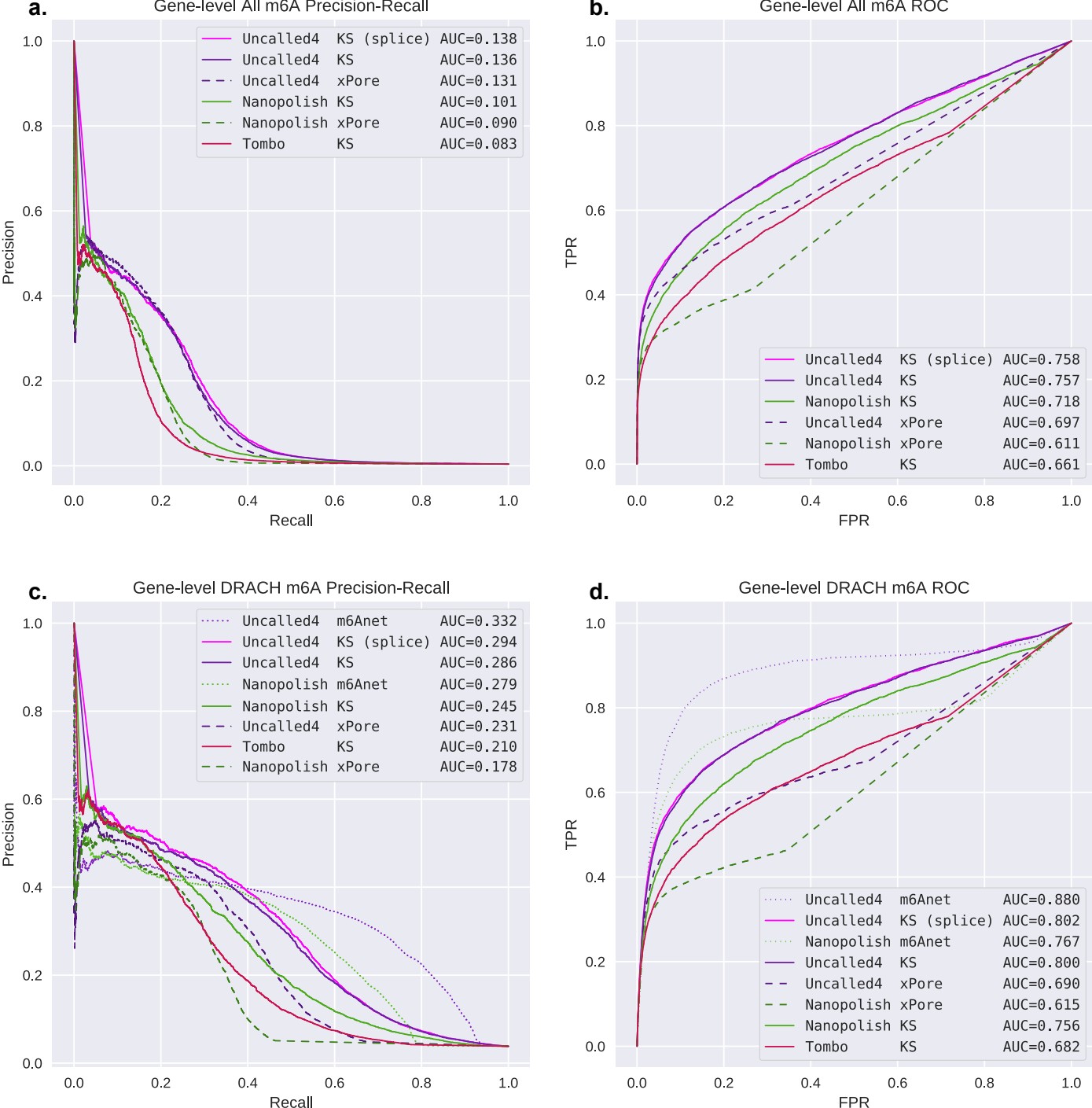

**Extended Data Fig. 8 | Gene-level comparative m6A detection.** Precision recall and ROC curves for gene-level comparative m6A detection in HEK293t in all contexts (**a-b**) and limited to DRACH sites (**c-d**). 'splice' indicates Uncalled4 spliced genome alignment. All other methods used transcriptome alignments with all multi-mappers included, averaged to the gene-level.

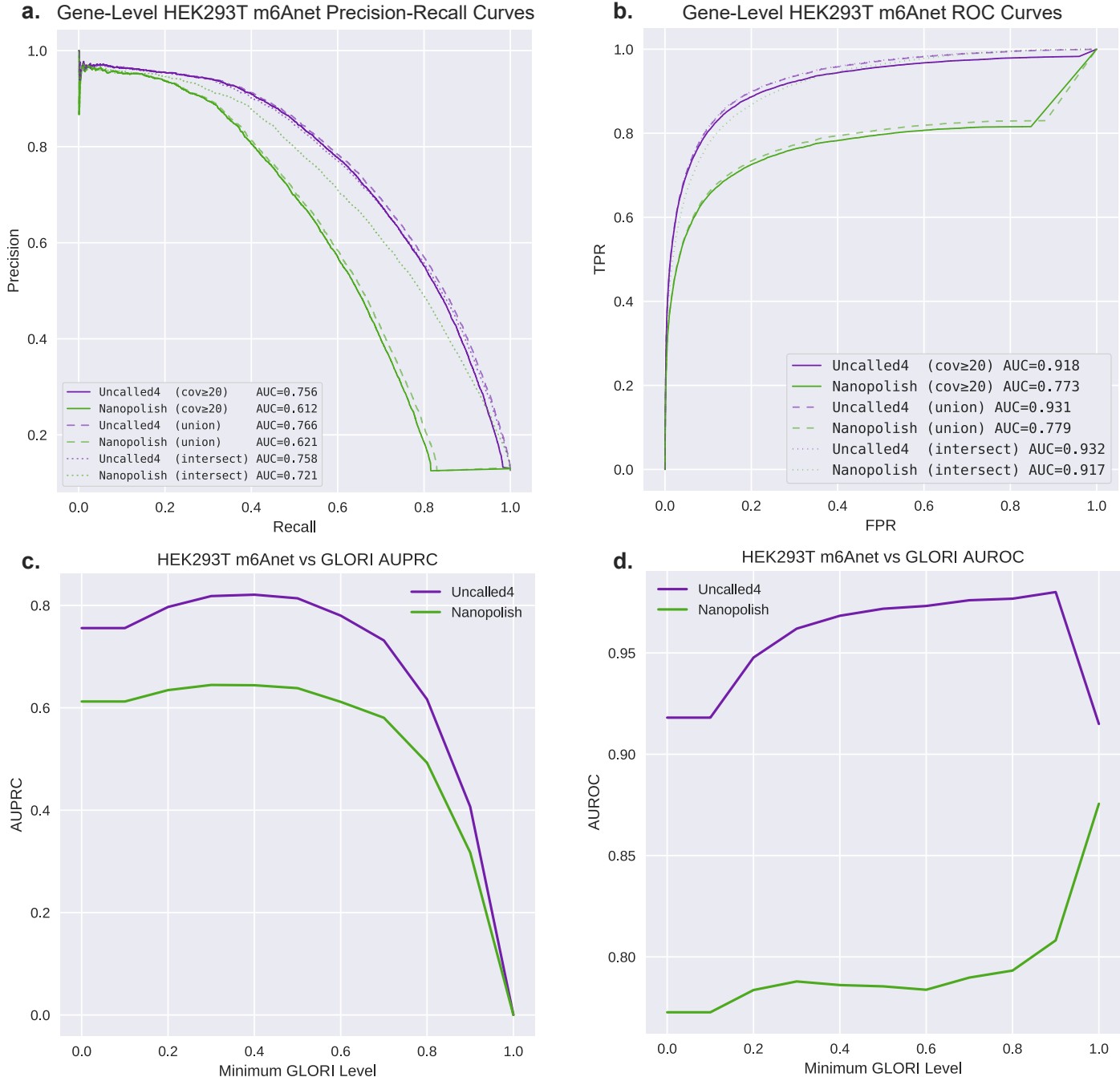

**Extended Data Fig. 9 | Gene-level HEK293T m6Anet.** Gene-level HEK293T m6Anet calls via transcript-to-genome (t2g) averaging. **(a)** Precision-recall and **(b)** ROC curves using GLORI labels with no level threshold. **(c)** Areas under the precision-recall and **(d)** ROC curves using different thresholds on GLORI levels.

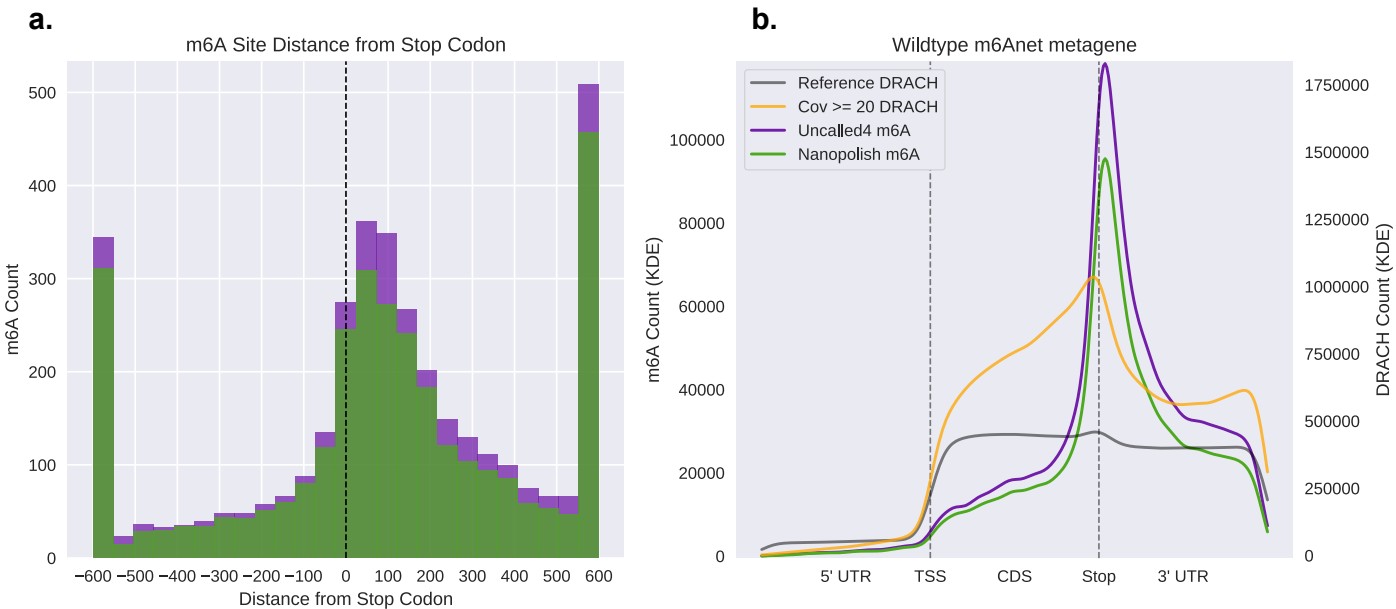

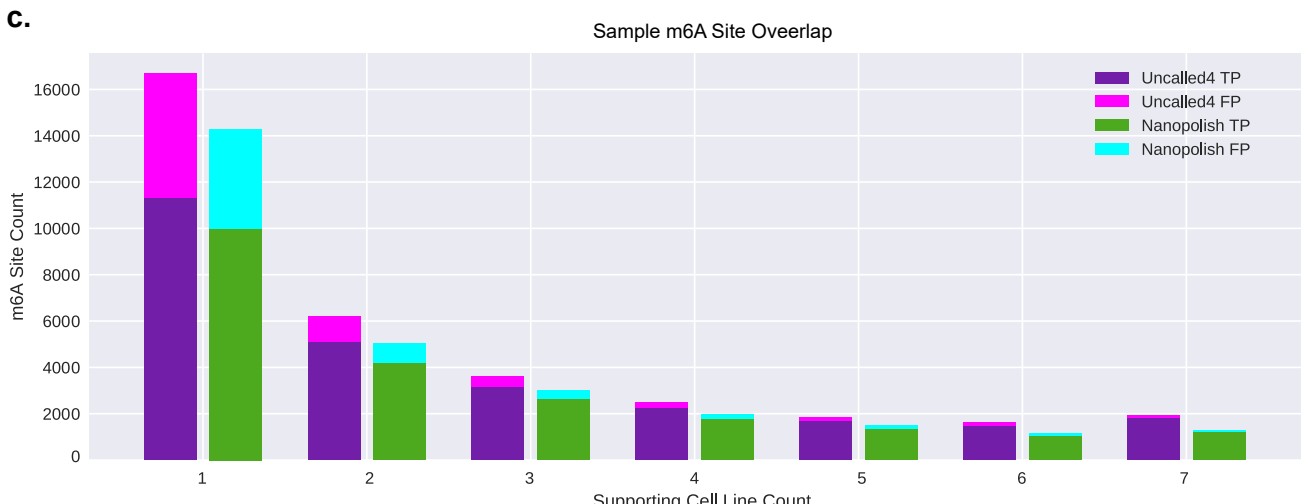

**Extended Data Fig. 10 | Cell line m6Anet analysis. (a)** Distance from annotated stop codon for transcript-level m6A sites found by Uncalled4 (purple) and Nanopolish (green) with m6Anet at matched 85% precision. **(b)** Metagene plot of the same m6A sites, with the distribution of reference DRACH sites (gray) and DRACH sites covered by the nanopore reads (orange). **(c)** Gene-level m6A counts by the number of cell lines they were found in, divided into putative 'true positives' (TP, in the m6A-Atlas) and putative 'false positives' (FP, missing from the m6A-Atlas).

# Reporting Summary

## Statistics

For all statistical analyses, confirm that the following items are present in the figure legend, table legend, main text, or Methods section.

| n/a | Confirmed | |
|---|---|---|
| ☐ | ☒ | The exact sample size (*n*) for each experimental group/condition, given as a discrete number and unit of measurement |
| ☒ | ☐ | A statement on whether measurements were taken from distinct samples or whether the same sample was measured repeatedly |
| ☐ | ☒ | The statistical test(s) used AND whether they are one- or two-sided<br>*Only common tests should be described solely by name; describe more complex techniques in the Methods section.* |
| ☒ | ☐ | A description of all covariates tested |
| ☒ | ☐ | A description of any assumptions or corrections, such as tests of normality and adjustment for multiple comparisons |
| ☐ | ☒ | A full description of the statistical parameters including central tendency (e.g. means) or other basic estimates (e.g. regression coefficient) AND variation (e.g. standard deviation) or associated estimates of uncertainty (e.g. confidence intervals) |
| ☒ | ☐ | For null hypothesis testing, the test statistic (e.g. *F*, *t*, *r*) with confidence intervals, effect sizes, degrees of freedom and *P* value noted<br>*Give P values as exact values whenever suitable.* |
| ☒ | ☐ | For Bayesian analysis, information on the choice of priors and Markov chain Monte Carlo settings |
| ☒ | ☐ | For hierarchical and complex designs, identification of the appropriate level for tests and full reporting of outcomes |
| ☐ | ☒ | Estimates of effect sizes (e.g. Cohen's *d*, Pearson's *r*), indicating how they were calculated |

*Our web collection on statistics for biologists contains articles on many of the points above.*

## Software and code

Policy information about availability of computer code

| Data collection | Sequencing was performed using Oxford Nanopore MinKNOW v5.2.8 |
|---|---|
| Data analysis | Basecalled using Guppy v6.4.8 for most data, except for Tombo where v4.0.1 was used. Basecalled alignments using  minimap2 v2.16. Signal alignments using Uncalled4 v4.0.0, Nanopolish v0.13.3, Tombo v1.5.1, and f5c v1.3. |

For manuscripts utilizing custom algorithms or software that are central to the research but not yet described in published literature, software must be made available to editors and reviewers. We strongly encourage code deposition in a community repository (e.g. GitHub). See the Nature Portfolio guidelines for submitting code & software for further information.

## Data

Policy information about availability of data

All manuscripts must include a data availability statement. This statement should provide the following information, where applicable:
- Accession codes, unique identifiers, or web links for publicly available datasets
- A description of any restrictions on data availability
- For clinical datasets or third party data, please ensure that the statement adheres to our policy

Drosphila sequencing data generated for this study is available at SRA bioproject PRJNA634693. All other datasets were obtained from publicly available sources. The E. coli rRNA data was obtained from [29] (SRA bioproject PRJNA634693). The constitutively incorporated BrdU dataset and matched control were from [30] (SRR8991355 and SRR8991351). The IVT HeLa cell direct RNA sequencing data used to train the RNA002 model was obtained from [53] (SRR23950400). HEK293T

## Human research participants

Policy information about studies involving human research participants and Sex and Gender in Research.

| Reporting on sex and gender | N/A |
|---|---|
| Population characteristics | N/A |
| Recruitment | N/A |
| Ethics oversight | N/A |

Note that full information on the approval of the study protocol must also be provided in the manuscript.

# Field-specific reporting

Please select the one below that is the best fit for your research. If you are not sure, read the appropriate sections before making your selection.

☒ Life sciences          ☐ Behavioural & social sciences          ☐ Ecological, evolutionary & environmental sciences

For a reference copy of the document with all sections, see nature.com/documents/nr-reporting-summary-flat.pdf

# Life sciences study design

All studies must disclose on these points even when the disclosure is negative.

| Sample size | Drosophila genomes were sequenced as this genome contains very possible 11-mer, allowing sufficient context to produce 9-mer k-mer models. |
|---|---|
| Data exclusions | We excluded Drosophila r10.4.1 260bps sequencing chemistry as this sequencing method was deprecated by ONT |
| Replication | All experiments were computational and can be replicated by running the command again |
| Randomization | Not relevant, no experimental groups required |
| Blinding | Not relevant, no experimental groups required |

# Reporting for specific materials, systems and methods

We require information from authors about some types of materials, experimental systems and methods used in many studies. Here, indicate whether each material, system or method listed is relevant to your study. If you are not sure if a list item applies to your research, read the appropriate section before selecting a response.

## Materials & experimental systems

| n/a | Involved in the study |
|---|---|
| ☒ ☐ | Antibodies |
| ☒ ☐ | Eukaryotic cell lines |
| ☒ ☐ | Palaeontology and archaeology |
| ☐ ☒ | Animals and other organisms |
| ☒ ☐ | Clinical data |
| ☒ ☐ | Dual use research of concern |

## Methods

| n/a | Involved in the study |
|---|---|
| ☒ ☐ | ChIP-seq |
| ☒ ☐ | Flow cytometry |
| ☒ ☐ | MRI-based neuroimaging |

# Animals and other research organisms

Policy information about studies involving animals; ARRIVE guidelines recommended for reporting animal research, and Sex and Gender in Research

| | |
|---|---|
| Laboratory animals | 15 newly eclosed (one day old) D. melanogaster males of the Oregon R strain (Bloomington stock center, #5) |
| Wild animals | Did not involve wild animals |
| Reporting on sex | Sex was not relevant to the study, but all individuals were male according to the stock facility |
| Field-collected samples | Did not involve field-collected samples |
| Ethics oversight | Not required for D. melanogaster |

Note that full information on the approval of the study protocol must also be provided in the manuscript.

