## [Peer Review File · Nature Methods]

Uncalled4 improves nanopore DNA and RNA modification detection via fast and accurate signal alignment

Corresponding Author: Dr Sam Kovaka

Version 0:

Decision Letter:

5th May 2024

Dear Dr Kovaka,

Your Article, "Uncalled4 improves nanopore DNA and RNA modification detection via fast and accurate signal alignment", has now been seen by 3 reviewers. As you will see from their comments below, although the reviewers find your work of potential interest, they have raised a number of concerns. We are interested in the possibility of publishing your paper in Nature Methods, but would like to consider your response to these concerns before we reach a final decision on publication.

We therefore invite you to revise your manuscript to fully address all these concerns.

Link Redacted

We hope to receive your revised paper within 3 months. If you cannot send it within this time, please let us know. In this event, we will still be happy to reconsider your paper at a later date so long as nothing similar has been accepted for publication at Nature Methods or published elsewhere.

OPEN SCIENCE REQUIREMENTS

REPORTING SUMMARY AND EDITORIAL POLICY CHECKLISTS

DATA AVAILABILITY

All novel DNA and RNA sequencing data, protein sequences, genetic polymorphisms, linked genotype and phenotype data, gene expression data, macromolecular structures, and proteomics data must be deposited in a publicly accessible database, and accession codes and associated hyperlinks must be provided in the "Data Availability" section.

CODE AVAILABILITY

Please include a "Code Availability" subsection in the Online Methods which details how your custom code is made available. Only in rare cases (where code is not central to the main conclusions of the paper) is the statement "available upon request" allowed (and reasons should be specified).

MATERIALS AVAILABILITY

ORCID

Nature Methods is committed to improving transparency in authorship. As part of our efforts in this direction, we are now requesting that all authors identified as 'corresponding author' on published papers create and link their Open Researcher and Contributor Identifier (ORCID) with their account on the Manuscript Tracking System (MTS), prior to acceptance. This applies to primary research papers only. ORCID helps the scientific community achieve unambiguous attribution of all scholarly contributions. You can create and link your ORCID from the home page of the MTS by clicking on 'Modify my Springer Nature

account'. For more information please visit www.springernature.com/orcid.

Sincerely,

Lin Tang, PhD
Senior Editor
Nature Methods

Reviewers' Comments:

Reviewer #1:

Remarks to the Author:

In the submitted manuscript, the authors present a computational method (Uncalled4) which aligns the raw signal from a nanopore sequencing experiment to the reference sequence to facilitate the discovery of nucleotide modifications from native DNA and RNA sequencing. The signal alignment algorithm uses information from the basecaller (moves) to guide a banded alignment. The authors compare their method against existing methods and demonstrate that their approach reduces computational resource requirements while increasing the accuracy and the total number of sites that can be detected. The signal alignment algorithm is complemented with additional tools for analyzing and visualizing the signal data.

The manuscript addresses an important problem which is essential to the detection of base modifications from nanopore sequencing data beyond those modifications that are detected by the basecalling algorithm. The manuscript is clearly written, and the method is comprehensively evaluated. The implementation seems to be well done, with detailed documentation, and numerous features that are helpful for processing and analyzing nanopore signal data. Please find my comments below

- The authors show the comparison of Uncalled4 with Nanopolish as an example (Fig 1c), and they provide the statistics from the comparison as a supplementary table. An additional comparison that could be insightful would be a comparison of signal statistics for each kmer after alignment. For example, I would expect that a cleaner alignment results in a reduction in standard deviation, is that the case?

- The KS test is used to compare the signal data to identify modified bases (for example to compare f5c and Uncalled4). The KS test is currently not well described, I assume this is used to compare the distribution of the mean signal from all reads at one site from one sample with the corresponding mean signal distribution from another sample? The description could be improved in the main text and methods section

- Figure 3b, the mean KS statistic seems to be higher for Uncalled4 compared to f5c across all positions, why would that be the case? It could be helpful for the reader to show the underlying distribution of the signal data at CpG sites to better illustrate why Uncalled4 performs better. E.g. is the data better separated? Do the signal distributions have lower standard deviation?

- the authors used the difference in modification rate multiplied by the WT modification probability as a score for m6Anet. Unlike the other methods, m6Anet can identify m6A from just one sample (without the comparison required by the KS test and xPore). The authors could illustrate the impact of using Uncalled4 vs Nanopolish in this scenario as well.

- How do the ROC curve or PR curve look for m6A when the difference between the modification rates is used without the WT modification probability? Does the difference in alignment method impact the significance score (WT modification probability) or the estimate of the difference in modification rate?

- "Interestingly, Nanopolish+m6Anet has a lower AUROC than Uncalled4+KS statistics, and has nearly the same AUPRC (only 0.02 higher), demonstrating that the alignment method has a larger impact than the downstream detection method in this context." While I understand what the authors want to say, the observation that "Nanopolish+m6Anet has a lower AUROC than Uncalled4+KS statistics" does not support the claim "that the alignment method has a larger impact than the downstream detection method in this context". For example, if the KS test performed better than m6Anet, this claim would not be valid, as the KS test could explain the difference. To make this claim in the main text, the authors should compare the impact of the alignment method (e.g. Nanopolish-m6Anet vs Uncalled4 m6Anet, maybe as % improvement) against the impact of the method (Uncalled4 m6Anet vs Uncalled 4 KS). I expect that the results will be similar, but the way it is currently written, the observation does not support this claim.

- the example of ABL is interesting. It looks like there are only very few reads supporting the fusion transcripts, otherwise the authors could compare m6A sites between the fusion and non-fusion transcript by including the fusion transcript in the alignment step.

- several figures are not referenced in the main text, and (without reading the methods) it is unclear why they are shown (e.g. Fig 1d,e,f)

- f5c provides many advantages over nanopolish, and it is actively maintained. While the alignment method is similar to Nanopolish, I would still recommend to mention f5c in the introduction as well.

- Figure 2b, the two distances (signal Jaccard distance and the signal-to-reference distance) are introduced as a way to compare two methods, but this figure only shows uncalled4. It is not clear what the distances shows in this figure. The legend suggests a comparison with "Native DTW", however, this is not clear from the text or legend.

-figure 4g and h, the label for Nanopolish and Uncalled4 sites seems to be swapped (Nanopolish has many more sites in these figures, also colour labels suggest a label error here)

Remarks on code availability

The code seems to be of high quality

Reviewer #2:

Remarks to the Author:

Uncalled4 is tackling an important problem in ONT signal processing: signal segmentation and alignment. It shows several advantages over existing methods, such as Nanopolish/f5c and Tombo, in terms of ease of use and the ability to process POD5 files directly, as well as the accuracy of signal alignment. It also provides a convenient signal processing method that can be readily used in downstream analysis in the form of BAM output without the need to share raw signal files. However, the authors need to show convincing evidence of the utility of their method over ONT's own move table output. Some comments are below to help improve the manuscript.

The authors correctly identify the problem with lower resolution of ONT move tables in terms of fixed block strides. However, recently developed modification detection tools (including but not limited to Rockfish and DeepMod2) have successfully used Guppy/Dorado move tables to align basecalls to reference for the purpose of DNA and RNA modification detection. The authors show how their signal alignment method is better than other methods at following the projection of ONT move table onto the reference sequence. But they still need to show that this provides better signal segmentation, fewer skips and higher resolution than ONT's move table, or that using Uncalled4 signal alignment provides better performance than ONT move table in terms of modification detection in downstream analysis. The paper only presents comparison with other signal alignment tools such as Nanopolish/f5c and Tombo, but does not make a strong case for whether the extra computational resources spent in running Uncalled4 will be beneficial over using just ONT move table directly. This is a key issue, since ultimately the method needs to yield some benefits to users in real data analysis to justify its usage.

The methods section is very light on the details of 1) signal preprocessing, 2) bcDTW algorithm, and 3) interpretation of BAM tags in terms of skip/stay, segmentation around indels and splicing. In terms of signal preprocessing, please provide more details of how "the event current means are iteratively normalized to correct for systematic deviation from the pore mode". Signal normalization has a huge impact on downstream tasks and various methods have been proposed and used. It is not clear what the two iterations of alignments are and how the linear regression is utilized. For bcDTW, more details of the algorithm and a brief visual guide that shows how an example ONT move table and a short CIGAR string is segmented into events and transformed into "ref-moves" and aligned to reference signal via DTW. It should also be made clear how the reference is created for a given reference sequence. Lastly, a visual guide of how the various artifacts of signal alignment such as skip/stays and indels/splicing should be interpreted via BAM tags. This can follow the pattern of visual guide provided by f5c: <https://hasindu2008.github.io/f5c/docs/output#dna-example>. As of now, it is not clear what the various tag values would be around a small indel or how the original move table relates the final signal alignment. Without such clear description and illustrations, it would be difficult for the broader community to use Uncalled4 signal alignment framework with confidence.

The signal alignment proposed by the authors assigns raw signal samples to aligned reference bases. One rationale for this can be that the basecalling may be error prone and the reference provides the correct underlying sequence. However, it is not clear how accurate the signal alignment is in the presence of true SNVs (that is, those positions that differ from the reference genome). More importantly, there is a case for aligning the raw signal to the actual basecalled sequence itself, especially now that the accuracy of basecalling has increased tremendously to the neighborhood of 1% with R10 flowcell with v14 chemistry. This is especially important because large scale rearrangements and SVs may not allow alignment to a reference genome, however it can still be important to analyze the signal of unaligned bases, which can later be aligned to a different reference genome or be analyzed by themselves for modifications. Signal to basecall alignment is provided in f5c (<https://hasindu2008.github.io/f5c/docs/output#dna-example>) and also used by DeepMod2 to elucidate signal characteristics within unaligned read segments. Authors should justify why they decided to leave out signal-to-basecall alignment that does not require a reference sequence in Uncalled4? In fact, implementation of such feature will be very useful for downstream tasks including DNA/RNA modification detection.

Can the authors explain whether Uncalled4 can be used to align signal or train pore models for unknown or synthetic modification datasets? For example, DNAscent and Forkseq use samples with BrdU replacing thymidine, often replacing up to 90% of thymidine. Such thymidine analogs may not be seen during basecaller training and thus the signal for analogs will be very different from the signal of standard nucleotides. How can Uncalled4 be used to align signals for samples with such synthetic modifications? Would it require re-training a pore model? The DNAscent/DNAscent2 data sets are publicly available and can be tested.

The benchmarking section on m6A is problematic, even though they have acknowledged it as well. They used HEK293T with over 100,000 gene-level sites in m6A-atlas. The current results from m6A-atlas on HEK293T come from many different sources

with vastly different sensitivity and coverage, and sometimes with little overlap between sources (even though the exact same cell line was assayed). Furthermore, most of these 100K sites are MeRIP-seq based results and they are simply not reliable. I do not think it is a good idea to take m6A-atlas at face value and treat it as a gold standard for any benchmarking studies. The authors are better off using a single m6A set, such as those inferred from GLORI, for the benchmarking and comparisons.

Furthermore, the statement that “we call each positive modification site a putative “true positive” (TP) if it occurs in any sample in the m6A-Atlas, acknowledging that this is an imperfect truth set and accuracy will vary between cell lines” is also quite problematic even though they also explicitly acknowledged this fact. Due to the nature of m6A-Atlas, which collects m6A calls from a diverse range of sources with varying quality, simply treating sites that appear in the database (in any cell) as true positive for a specific cell line is not a good idea, otherwise all cell lines have identical m6A patterns in data analysis.

They analyzed or trained models with in vitro transcribed (IVT) from HeLA and E coli. The IVT for other cell lines can be used to assess false positives of calling RNA modifications since IVT samples should not have any modifications. This is another important point to address in light of the concerns above on over-calling “positive” sites.

Remarks on code availability

More documentation is necessary to show how to use the software. For example, what is a pore model (the collection of preset models) and how to specify a custom pore model in a TSV file.

Reviewer #3:

Remarks to the Author:

The authors, Sam Kovaka et. al. described a new nanopore signal alignment tool called, Uncalled4. This tool is designed to align the raw signal data from nanopore sequencers to a reference sequence, which is a fundamental step for any analysis using the nanopore raw signal. Uncalled4 is an impressive tool kit which represents a major improvement in nanopore signal alignment. The authors show that the dynamic time warping signal alignment method implemented within Uncalled4 increases the number of aligned segments and more accurately represents the nucleotide polymers that are measured by the nanopore sensors. The authors focus on how the signal alignment step can be used for a variety of analysis steps, such as inferring a pore model, and detecting nucleotide modifications. They clearly show how fundamental the signal alignment step is for modification detection, by testing the most widely used signal alignment tool, Nanopolish, and Uncalled4 as the data preprocessing step for RNA modification detection tools. When compared against annotated m6A sites, the modification detection software tools had higher scores when using Uncalled4 as the signal alignment tool instead of Nanopolish or Tombo. Despite showing an impressive tool, I have some points that I'd like the authors to clarify.

Major points:

1. Figure 1 comments: Panel 1B: The Jaccard distance is defined in the text as “the fraction of raw samples that both methods aligned to the same position”. However, in panel B, it appears that only one method is being displayed. How is the Jaccard distance being used in this panel? Panel C: legend description doesn't explain what the blue line (compare) is representing, more details here would help to better understanding the plot. Panel D: the label for the heatmap color bar appears to be cropped. Panel D: the red arrows are not labeled in the plot, nor the figure legend. I see what they are trying to depict, but I find them largely distracting and think that describing in the figure legend can provide the necessary context for the other figure panels. Panel E: Including the Native and IVT legend in the plot would help for understanding the plot immediately.

2. In this sentence: “Nanopolish has slightly lower average Jaccard distance in r9.4.1 DNA and RNA, with less than 50% overlap on average for all aligners and less than 2% difference between Uncalled4 and Nanopolish” it appears that the Jaccard distance can be calculated individually for each tool, but in the text the Jaccard distance is defined as a comparative method. I am not super familiar with the Jaccard distance, but it is my understanding that it is a comparative statistic. Can the text be clarified what sets are being compared to calculate the Jaccard distance here and in other parts of the manuscript, such as in figure 1B?

3. Do the authors expect the file size and computation time to scale linearly as data sets increase by 4000 reads?

4. The color coding in Figure 4 e&f is not entirely clear. The negative values (green) suggest that these are m6A counts found by m6A net using Nanopolish as the signal alignment method, but not by m6A net using Uncalled4 as the signal aligner. But does this example mean that no m6A sites were identified by m6A net using Uncalled4 or simply X many more using Nanopolish?

5. In describing the dynamic time warping algorithm, much care was given to the alignment space banding, but a description of the distance equation was not included in the methods section.

6. The two implemented data scaling methods are not entirely clear, what is the difference between reference-guided method-of-moments scaling and model-guided methods-of-moments scaling?

Minor points:

1. The last sentence of first paragraph in the introduction is a little unclear what the authors mean by “although only a few have been detected thus far”. It seems at odds with the first half of the sentence that there are 150 known RNA modifications. Do the authors mean instead that only a few have been detected using nanopore technology?

“Over 150 known RNA modifications are known to exist, although only a few have been detected thus far, with varying accuracy [10].”

2. I think that the right parenthesis should be moved in this sentence to only include the acronym “such as Dynamic Time Warping (DTW, used by Tombo) or Hidden Markov Models (HMMs, used by Nanopolish)” should be “such as Dynamic Time Warping (DTW), as used by Tombo or Hidden Markov Models (HMMs), as used by Nanopolish”.
3. Figure 1 panels B and C, is there a reason that they are different regions of the E. coli 16s rRNA?
4. Throughout the text the authors use both 5mCpG and 5mC. I think it would be better to pick one naming convention where possible. The authors use 5mCpG six times and 5mC seven times.
5. Table 1: It is my understanding that table legends should appear above the table instead of below like with figures.
6. Supplemental figure 3e would help to include the color coding within the plot instead of leaving the color coding just to the figure legend.
7. Supplemental figure 4 a&b would benefit from including the central kmer base color coding within at least one of these two plots.
8. Supplemental figure 10 appears to have a legend and figure title mismatch. Also, the black dots in the plot are not described anywhere.

Remarks on code availability

I made a fresh install of gcc and a clean conda virtual environment with python version 3.8, and tried to install uncalled4 using the github method. I was unable to install uncalled4. Here is hopefully a useful error code for the authors:

In file included from src/cpp/config.cpp:1:

In file included from ./src/cpp/config.hpp:30:

In file included from ./src/cpp/dtw.hpp:8:

In file included from ./src/cpp/seq.hpp:32:

In file included from ./src/cpp/pore_model.hpp:37:

./src/cpp/intervals.hpp:167:9: error: 'auto' deduced as 'std::vector<interval<long long>::const_iterator' (aka ' __wrap_iter<const Interval<long long> *>') in declaration of 'a' and deduced as 'std::vector<interval<long long>::iterator' (aka ' __wrap_iter<interval<long long> *>') in declaration of 'b'

auto a = coords.begin(),

^ ~~~~~

./src/cpp/aln.hpp:175:25: note: in instantiation of member function 'IntervalIndex<long long>::intersect' requested here

index = a.index.intersect(b.index);

^

26 warnings and 6 errors generated.

error: command '/usr/bin/clang' failed with exit code 1

[end of output]

note: This error originates from a subprocess, and is likely not a problem with pip.

ERROR: Failed building wheel for uncalled4

Running setup.py clean for uncalled4

Failed to build uncalled4

ERROR: Could not build wheels for uncalled4, which is required to install pyproject.toml-based projects

However, independent of this review, one of my students was able to install uncalled4 on both his local machine and on the HPC. He has begun testing uncalled4 in our RNA modification detection framework. The code executed as expected in his hands.

The github documentation is clear (aside from the normalization comment mentioned above). The code is very readable, but there are several areas where the code was commented out, likely because those lines are no longer necessary. Also, it was a little difficult to follow the logic on how the different subcommands called the different functions when I was tried to track down the code for the convert subcommand.

Version 1:

Decision Letter:

Our ref: NMETH-A55766A

23rd Sep 2024

Dear Dr. Kovaka,

Thank you for submitting your revised manuscript "Uncalled4 improves nanopore DNA and RNA modification detection via fast and accurate signal alignment" (NMETH-A55766A). It has now been seen by the original referees and their comments are below. The reviewers find that the paper has improved in revision, and therefore we'll be happy in principle to publish it in Nature Methods, pending minor revisions to satisfy the referees' final requests and to comply with our editorial and formatting

guidelines.

TRANSPARENT PEER REVIEW

ORCID

Sincerely,

Lin Tang, PhD
Senior Editor
Nature Methods

Reviewer #1 (Remarks to the Author):

The authors have addressed my comments. I don't have further comments

Reviewer #1 (Remarks on code availability):

the code is well documented and of high quality

Reviewer #2 (Remarks to the Author):

The authors have made an effort to address my previous comments. The additional analysis on IVT is especially interesting (and I do think it is very important). The use of additional "gold standard" other than or in addition to m6A-atlas is also important to assess the performance of the new approach. I think uncalled4 could be a very useful tool to the community doing signal-level analysis of nanopore data and recommend the publication of this manuscript.

Reviewer #2 (Remarks on code availability):

It would be better if they have some example output (including figures from dotplot and trackplot), possibly in the example/ folder, so that users know what are expected and can compare results to ensure the correct execution of the tool.

Reviewer #3 (Remarks to the Author):

The authors of uncalled4 have made many improvements to the manuscript and github pages since their initial submission. The added clarity on how the different pore models could be trained, and demonstrating the utility of a custom pore model for BrdU detection was a beneficial addition. Furthermore, rearranging figure 1 greatly improved the clarity on how the uncalled4 DTW alignment compared with the move table and Nanopolish alignments. Additionally, the expanded methods section is much clearer. Splitting the RNA modification detection section made the logic and analysis easier to follow. In this updated manuscript, the authors have addressed all my previous comments, but I still have some minor comments about figure 4.

In figure 4, the authors use m6ACE-seq [<http://dx.doi.org/10.1038/s41587-021-00949-w>] as their ground truth set for panel A, but then use the GLORI as their ground truth set for panels B and C. As all the data in Figure 4 is from HEK293 cells, it seems like it would be more appropriate to use the same ground truth set for AUPRC calculations. Given that the AUPRC values in figure 4a range between 0.08 and 0.35, yet in figure 4b they range between 0.578 and 0.744, the m6ACE-seq seem ill suited as a ground truth set for m6A detection.

The claim about gene-level modification calls being more robust than transcript-level could simply be explained by increased power from more observations for the statistical tests at each reference position than anything to do with ambiguous mapping to transcripts. And the AUPRC values for the two uncalled4 based alignments (the only tool that is appropriate for gene-level alignments) show only a modest improvement when switching to gene-level modification detection. Have the authors tried gene-level modification detection after down sampling to the average number of reads aligned to individual transcripts? Furthermore, it would make comparisons between transcript-level and gene-level easier if the y-axis ranges were identical for the All context and the DRACH context columns.

The claim "We also computed KS statistics using uncorrected ref-moves and found it performed worse than KS statistics using any other alignment method" was not supported by any quantitative evidence in the prose, nor any figure or supplemental figure in the manuscript.

Reviewer #3 (Remarks on code availability):

The code base is organized really well and the improvements to the code since the initial submission have made it much more clear and easy to understand how the different parts work.

Version 2:

Decision Letter:

16th Feb 2025

Dear Dr Kovaka,

Thank you very much for sending us the updated version of your Article "Uncalled4 improves nanopore DNA and RNA modification detection via fast and accurate signal alignment". I am pleased to inform you that this paper has now been accepted for publication in Nature Methods. The received and accepted dates will be 15th Mar 2024 and 16th Feb 2025. This note is intended to let you know what to expect from us over the next month or so, and to let you know where to address any further questions.

Over the next few weeks, your paper will be copyedited to ensure that it conforms to Nature Methods style. Once your paper is typeset, you will receive an email with a link to choose the appropriate publishing options for your paper and our Author Services team will be in touch regarding any additional information that may be required. It is extremely important that you let us know now whether you will be difficult to contact over the next month. If this is the case, we ask that you send us the contact information (email, phone and fax) of someone who will be able to check the proofs and deal with any last-minute problems.

Please feel free to contact me if you have questions about any of these points. Thank you again for publishing your paper at Nature Methods!

Best regards,

Lin Tang, PhD
Senior Editor
Nature Methods

** Visit the Springer Nature Editorial and Publishing website at http://editorial-jobs.springernature.com?utm_source=ejP_NMeth_email&utm_medium=ejP_NMeth_email&utm_campaign=ejp_Nmeth for more information about our career opportunities. If you have any questions please click [here](mailto:editorial.publishing.jobs@springernature.com).

Open Access This Peer Review File is licensed under a Creative Commons Attribution 4.0 International License, which permits use, sharing, adaptation, distribution and reproduction in any medium or format, as long as you give appropriate credit to the original author(s) and the source, provide a link to the Creative Commons license, and indicate if changes were made. In cases where reviewers are anonymous, credit should be given to 'Anonymous Referee' and the source.

Dear Dr. Tang,

We thank the reviewers for their many helpful comments. We have made several major changes and enhancements based on their suggestions. At the suggestion of the reviewers, two new alignment modes have been added to Uncalled4: signal-to-read alignment, which performs reference-free alignment to the basecalled read sequence, and uncorrected *read-moves* and *ref-moves* output. These approaches are discussed in the first results section, where we note that signal-to-read alignment is a promising approach but that uncorrected moves are far less accurate than Uncalled4 alignments. We also split the HEK293T RNA modification detection results into two sections, comparative and m6Anet, and used the new more accurate short-read m6A detection method named GLORI to measure accuracy. We also trained a model for BrdU with Uncalled4, and demonstrate Uncalled4's ability to accurately detect these nucleotide analogs. We also made several other minor changes directly addressing reviewer comments, described in detail below.

In addition to addressing the reviewer comments, we also added two additional analyses to highlight the latest pore chemistry from Oxford Nanopore (RNA004 sequencing data) and a related analysis of the dwell time. We previously briefly mentioned RNA004 and analyzed its pore model in the original manuscript, but since our initial submission a HEK293T RNA004 dataset has been made available, allowing us to add Uncalled4 and f5c alignment statistics to **Table 1** and **Figure 2**. We note that the RNA004 pore model is highly similar to RNA001/RNA002, so any increase in accuracy would be due to more consistent sequencing speed or other changes to sequencing chemistry to reduce noise. Exploration of this led to an interesting result on dwell time, where we find that the dwell time is affected by the sequence in the pore and the sequence upstream near the motor protein. Similar results had previously been noted in RNA (Stephenson et al. 2020), but to our knowledge this is the first thorough analysis of position-specific dwell time effects in all major ONT sequencing methods. Two panels have been added to **Figure 2**, and the results are discussed in "Read signal and pore model characteristics".

We have also updated the Uncalled4 software to version 4.1, which includes support for RNA004 alignment, along with several additions requested by the reviewers: scripts for BAM parsing, pore model conversion, and dwell time analysis; changes to the command line arguments to support signal-to-read alignment and *moves* output; adjustment of r10.4 k-mer coordinates so that the reported positions are defined relative to the central bases. We also improved KS statistic transcript-to-gene averaging by filtering out transcript-level sites under 20x coverage to be consistent with xPore and m6Anet, where previously low-coverage transcripts were included and only filtered out based on average gene-level coverage. Importantly, we ensured that the signal

alignment algorithm produces the same results presented in the originally submitted manuscript, only the gene-level results changed slightly due to the change in transcript-level filtering, and all other changes only affect usability and input/output formats.

Overall, these updates strengthen our previous submission and we appreciate all of the helpful suggestions from the reviewers. Below are our point-by-point responses to the reviewers, and we have highlighted our changes in the revised manuscript.

Thank you for your consideration,

Sam Kovaka (on behalf of all the authors)

~~~~~

**Reviewer #1:**

**Remarks to the Author:**

**In the submitted manuscript, the authors present a computational method (Uncalled4) which aligns the raw signal from a nanopore sequencing experiment to the reference sequence to facilitate the discovery of nucleotide modifications from native DNA and RNA sequencing. The signal alignment algorithm uses information from the basecaller (moves) to guide a banded alignment. The authors compare their method against existing methods and demonstrate that their approach reduces computational resource requirements while increasing the accuracy and the total number of sites that can be detected. The signal alignment algorithm is complemented with additional tools for analyzing and visualizing the signal data.**

**The manuscript addresses an important problem which is essential to the detection of base modifications from nanopore sequencing data beyond those modifications that are detected by the basecalling algorithm. The manuscript is clearly written, and the method is comprehensively evaluated. The implementation seems to be well done, with detailed documentation, and numerous features that are helpful for processing and analyzing nanopore signal data. Please find my comments below**

Thank you for your enthusiastic comments and suggestions.

**- The authors show the comparison of Uncalled4 with Nanopolish as an example (Fig 1c), and they provide the statistics from the comparison as a supplementary table. An additional comparison that could be insightful would be a comparison of signal statistics for each kmer after alignment. For example, I would expect**

**that a cleaner alignment results in a reduction in standard deviation, is that the case?**

Thank you for the suggestion. Current-level statistics are definitely useful and informative, and we have added a column to Table 1 measuring the median absolute difference (MAD) between per-read-k-mer mean currents and pore model predictions for each aligner and sequencing chemistry. We did not compare per-k-mer current standard deviations because by default Tombo does not output those values, and Nanopolish only reports per-event standard deviations that can not be accurately converted to per-k-mer values. We also note that Nanopolish and Tombo filter out many sites and reads, respectively, which likely contributes to the reduced model MAD, but also reduces their coverage and thus sensitivity to nucleotide modifications (see **Table 1**). We expanded the first results section to discuss these results, and the statistics are also plotted in **Supplemental Figure 1c** and **Supplemental Figure 3a**.

**- The KS test is used to compare the signal data to identify modified bases (for example to compare f5c and Uncalled4). The KS test is currently not well described, I assume this is used to compare the distribution of the mean signal from all reads at one site from one sample with the corresponding mean signal distribution from another sample? The description could be improved in the main text and methods section**

Your interpretation of the KS test is corrected. To clarify this point, we have expanded our description of KS statistics in the main text to clarify this under “DNA modification model training and detection” and briefly in “Comparative RNA modification detection”, and in the methods under “Modification Detection, Training, and Assessment”

**- Figure 3b, the mean KS statistic seems to be higher for Uncalled4 compared to f5c across all positions, why would that be the case? It could be helpful for the reader to show the underlying distribution of the signal data at CpG sites to better illustrate why Uncalled4 performs better. E.g. is the data better separated? Do the signal distributions have lower standard deviation?**

Thank you for pointing this out. The increased background KS statistic for Uncalled4 was caused by erroneously using the Uncalled4-trained r10.4 model for Uncalled4 alignment and the default model for f5c. We revised the main figure such that both aligners use the same f5c model, and added **Supplemental Figure 9d** to show the effect of using each model with each aligner. We greatly appreciate the reviewers pointing out this discrepancy.

We also added a panel to the main **Figure 3** to display the z-score alongside, which shows the direction and magnitude of change in current. This shows that on average 5mCpG causes a small reduction in current before a larger increase at the central position, followed by another small drop, which the KS statistics did not show. We thank the reviewer for the suggestion.

**- the authors used the difference in modification rate multiplied by the WT modification probability as a score for m6Anet. Unlike the other methods, m6Anet can identify m6A from just one sample (without the comparison required by the KS test and xPore). The authors could illustrate the impact of using Uncalled4 vs Nanopolish in this scenario as well.**

The single-sample m6Anet results were previously introduced after the comparative results, with only a brief mention of HEK293T results before moving on to more cell lines. We have now split the HEK293T results into comparative “Comparative RNA modification detection” and “HEK293T RNA modification detection with m6Anet” and added a main figure that highlights both approaches on HEK293T (**Figure 4**). We hope this is a more clear presentation of the results.

**- How do the ROC curve or PR curve look for m6A when the difference between the modification rates is used without the WT modification probability? Does the difference in alignment method impact the significance score (WT modification probability) or the estimate of the difference in modification rate?**

We added **Supplemental Figure 18** to compare different statistics for comparing m6Anet results between multiple samples, showing that the difference in modification rate multiplied by the WT probability generally performs best. It is notable that excluding the probability performs slightly better for Nanopolish, but the difference is minor and the comparison to Uncalled4 remains unchanged. We thank the reviewer for suggesting this useful addition.

**- "Interestingly, Nanopolish+m6Anet has a lower AUROC than Uncalled4+KS statistics, and has nearly the same AUPRC (only 0.02 higher), demonstrating that the alignment method has a larger impact than the downstream detection method in this context." While I understand what the authors want to say, the observation that "Nanopolish+m6Anet has a lower AUROC than Uncalled4+KS statistics" does not support the claim "that the alignment method has a larger impact than the downstream detection method in this context". For example, if the KS test performed better than m6Anet, this claim would not be valid, as the KS test could explain the difference. To make this claim in the main text, the authors should compare the impact of the alignment method (e.g. Nanopolish-m6Anet vs Uncalled4 m6Anet, maybe as % improvement) against the impact of the method (Uncalled4 m6Anet vs Uncalled 4 KS). I expect that the results will be similar, but the way it is currently written, the observation does not support this claim.**

We revised that sentence to better support our claim, and moved the relevant results from the supplement to the main **Figure 4**, demonstrating that m6Anet does perform better when the same aligner is used, but the use of Uncalled4 makes a larger difference in this scenario

**- the example of ABL is interesting. It looks like there are only very few reads supporting the fusion transcripts, otherwise the authors could compare m6A**

**sites between the fusion and non-fusion transcript by including the fusion transcript in the alignment step.**

This is a very interesting idea, although unfortunately, there are too few reads supporting the *ABL1-BCR* fusion to perform such an analysis, even when combining all K562 replicates available from SG-Nex. *BCR* alone does not reach 20x coverage in any K562 replicate, and the fusion is included in fewer than 10 reads across all replicates. We will investigate this in a future project with additional RNA sequencing.

**- several figures are not referenced in the main text, and (without reading the methods) it is unclear why they are shown (e.g. Fig 1d,e,f)**

Apologies for the error, we rearranged and revised **Figure 1** and the opening “results” section so every panel is referenced in order. We also swapped the pore model plots in **Fig. 1a** and **Fig. 2a**, since we believe the sequence logos are a better introduction to pore model compositions.

**- f5C provides many advantages over nanopolish, and it is actively maintained. While the alignment method is similar to Nanopolish, I would still recommend to mention f5c in the introduction as well.**

Thank you for pointing this out. We have added a description of f5c to the introduction.

**- Figure 2b, the two distances (signal Jaccard distance and the signal-to-reference distance) are introduced as a way to compare two methods, but this figure only shows uncalled4. It is not clear what the distances shows in this figure. The legend suggests a comparison with "Native DTW", however, this is not clear from the text or legend.**

The originally included comparison statistics were measuring distance to *ref-moves*, but we realize that was not the most logical order to present those results and they were not described adequately. We removed distance measures from **Fig. 1a** and replaced them with the base-level layers that define a signal alignment: current, current\_sd, and dwell.

**-figure 4g and h, the label for Nanopolish and Uncalled4 sites seems to be swapped (Nanopolish has many more sites in these figures, also colour labels suggest a label error here)**

We have fixed those swapped labels and thank the reviewers for pointing out the mistake.

**Remarks on code availability**  
**The code seems to be of high quality**

Thank you!

---

Reviewer #2:

**Remarks to the Author:**

Uncalled4 is tackling an important problem in ONT signal processing: signal segmentation and alignment. It shows several advantages over existing methods, such as Nanopolish/f5c and Tombo, in terms of ease of use and the ability to process POD5 files directly, as well as the accuracy of signal alignment. It also provides a convenient signal processing method that can be readily used in downstream analysis in the form of BAM output without the need to share raw signal files. However, the authors need to show convincing evidence of the utility of their method over ONT's own move table output. Some comments are below to help improve the manuscript.

Thank you for the helpful comments and suggestions.

The authors correctly identify the problem with lower resolution of ONT move tables in terms of fixed block strides. However, recently developed modification detection tools (including but not limited to Rockfish and DeepMod2) have successfully used Guppy/Dorado move tables to align basecalls to reference for the purpose of DNA and RNA modification detection. The authors show how their signal alignment method is better than other methods at following the projection of ONT move table onto the reference sequence. But they still need to show that this provides better signal segmentation, fewer skips and higher resolution than ONT's move table, or that using Uncalled4 signal alignment provides better performance than ONT move table in terms of modification detection in downstream analysis. The paper only presents comparison with other signal alignment tools such as Nanopolish/f5c and Tombo, but does not make a strong case for whether the extra computational resources spent in running Uncalled4 will be beneficial over using just ONT move table directly. This is a key issue, since ultimately the method needs to yield some benefits to users in real data analysis to justify its usage.

Thank you for the comments. We had not adequately demonstrated the low-resolution of the basecaller *moves*. We have added a description of *moves* to the introduction, where we note that Rockfish and DeepMod2 use them to extract 21-31 base segments of signal, so an offset of 1-2 bases at the endpoints won't substantially impact the neural network classification. We also compare the difference between the per-read-k-mer mean currents with the pore model to show that both raw read-level *moves* and *ref-moves* form a bimodal distribution, while bcDTW is a much more normal distribution centered at zero, as expected (**Supplemental Figure 3a**).

We also tested the ability of *ref-moves* to detect RNA modifications by computing current-level KS statistics in the HEK293T METTL3 KO/WT comparison (**Supplemental Figure 12**). *Ref-moves* performed worse than all other aligners, although surprisingly the AUPRC is closer to Nanopolish or Tombo than those aligners are to Uncalled4.

The methods section is very light on the details of 1) signal preprocessing, 2) bcDTW algorithm, and 3) interpretation of BAM tags in terms of skip/stay, segmentation around indels and splicing.

Thank you for pointing this out. We have expanded each of these methods sections, and added a panel to **Supplemental Figure 2** to better illustrate bcDTW.

In terms of signal preprocessing, please provide more details of how "the event current means are iteratively normalized to correct for systematic deviation from the pore mode". Signal normalization has a huge impact on downstream tasks and various methods have been proposed and used. It is not clear what the two iterations of alignments are and how the linear regression is utilized.

We have expanded our description of the normalization methods, apologies for the previous lack of detail.

For bcDTW, more details of the algorithm and a brief visual guide that shows how an example ONT move table and a short CIGAR string is segmented into events and transformed into "ref-moves" and aligned to reference signal via DTW. It should also be made clear how the reference is created for a given reference sequence.

Lastly, a visual guide of how the various artifacts of signal alignment such as skip/stays and indels/splicing should be interpreted via BAM tags. This can follow the pattern of visual guide provided by f5c:

<https://hasindu2008.github.io/f5c/docs/output#dna-example>. As of now, it is not clear what the various tag values would be around a small indel or how the original move table relates the final signal alignment. Without such clear description and illustrations, it would be difficult for the broader community to use Uncalled4 signal alignment framework with confidence.

To address these comments, we added a panel to **Supplemental Figure 2** to illustrate a simple example of translation from *read-moves* to *ref-moves* via a minimap2 alignment, including the corresponding CIGAR string in the caption. We have also added more detail throughout the methods, particularly under "Alignment Encoding and Formats", to clarify how artifacts like skips are handled, and we show an example of a large deletion in **Supplemental Figure 3b**.

The signal alignment proposed by the authors assigns raw signal samples to aligned reference bases. One rationale for this can be that the basecalling may be error prone and the reference provides the correct underlying sequence. However, it is not clear how accurate the signal alignment is in the presence of true SNVs (that is, those positions that differ from the reference genome). More importantly, there is a case for aligning the raw signal to the actual basecalled sequence itself, especially now that the accuracy of basecalling has increased

tremendously to the neighborhood of 1% with R10 flowcell with v14 chemistry. This is especially important because large scale rearrangements and SVs may not allow alignment to a reference genome, however it can still be important to analyze the signal of unaligned bases, which can later be aligned to a different reference genome or be analyzed by themselves for modifications. Signal to basecall alignment is provided in f5c (<https://hasindu2008.github.io/f5c/docs/output#dna-example>) and also used by DeepMod2 to elucidate signal characteristics within unaligned read segments. Authors should justify why they decided to leave out signal-to-basecall alignment that does not require a reference sequence in Uncalled4? In fact, implementation of such feature will be very useful for downstream tasks including DNA/RNA modification detection.

We agree that signal-to-read alignment is a promising method for which Uncalled4 is well suited, so we have implemented it and added a description to the opening of the “results” section. We demonstrate potential uses and issues in **Supplemental Figure 3** by substituting or deleting bases from the *D. melanogaster* reference, causing errors in signal-to-reference alignment. We find that current levels in signal-to-read alignments are indeed more similar to the pore model, but also note that sequencing errors likely contribute to this since we expect little genetic variation in this sample. We thank the reviewer for their suggestion to implement signal-to-read alignment, and hope it will encourage development of reference-free modification detection methods.

**Can the authors explain whether Uncalled4 can be used to align signal or train pore models for unknown or synthetic modification datasets? For example, DNAscent and Forkseq use samples with BrdU replacing thymidine, often replacing up to 90% of thymidine. Such thymidine analogs may not be seen during basecaller training and thus the signal for analogs will be very different from the signal of standard nucleotides. How can Uncalled4 be used to align signals for samples with such synthetic modifications? Would it require re-training a pore model? The DNAscent/DNAscent2 data sets are publicly available and can be tested.**

Uncalled4 model training is well-suited for any sample where most bases in a particular context are modified, making BrdU an excellent application. To highlight this capability, we trained a BrdU pore model using the DNAscent2 dataset and described the results in the “DNA modification model training and detection” results (**Fig. 3c-d**). This dataset used the experimental r9.5 sequencing chemistry, which we found to be highly similar to r9.4. Unfortunately no r10.4 BrdU training datasets have been made available to our knowledge. We demonstrate a simple use for the Uncalled4 BrdU pore model, showing it improves BrdU k-mer classification using a simple model current difference metric, compared to using the DNAscent2 pore model for Uncalled4 alignment.

**The benchmarking section on m6A is problematic, even though they have acknowledged it as well. They used HEK293T with over 100,000 gene-level sites in**

**m6A-atlas. The current results from m6A-atlas on HEK293T come from many different sources with vastly different sensitivity and coverage, and sometimes with little overlap between sources (even though the exact same cell line was assayed). Furthermore, most of these 100K sites are MeRIP-seq based results and they are simply not reliable. I do not think it is a good idea to take m6A-atlas at face value and treat it as a gold standard for any benchmarking studies. The authors are better off using a single m6A set, such as those inferred from GLORI, for the benchmarking and comparisons.**

Thank you for the comment, we agree that relying on the m6A-Atlas was not ideal and greatly appreciate the suggestion to evaluate GLORI data. The use of the GLORI data dramatically improved the m6Anet AUPRC and AUROC for both Uncalled4 and Nanopolish, and further highlights the improved accuracy of Uncalled4 (**Fig. 4b, Supplemental Figure 17**). We added **Figure 4** to highlight the HEK293T RNA m6A results, and split the results section into “Comparative RNA modification detection” and “HEK293T RNA modification detection with m6Anet” to clarify the presentation.

**Furthermore, the statement that “we call each positive modification site a putative “true positive” (TP) if it occurs in any sample in the m6A-Atlas, acknowledging that this is an imperfect truth set and accuracy will vary between cell lines” is also quite problematic even though they also explicitly acknowledged this fact. Due to the nature of m6A-Atlas, which collects m6A calls from a diverse range of sources with varying quality, simply treating sites that appear in the database (in any cell) as true positive for a specific cell line is not a good idea, otherwise all cell lines have identical m6A patterns in data analysis.**

We agree the m6A-Atlas should not be treated as the set of “true positives”, and instead changed our phrasing to treat it as a set of possibly modified sites. We also added alternate m6A-Atlas filtering criteria by limiting to matched HEK293T, A549, and HepG2-specific sites to **Supplemental Figure 19**, showing that the relative performance of Uncalled4 and Nanopolish is similar to those estimated using GLORI labels, with the lower quality m6A-Atlas labels generally lowering AUCs for both aligners. We also experimented with excluding all immunoprecipitation-based assays and observed the same trends. We believe that opening with the GLORI labels as our “gold standard” helps establish that the m6A-Atlas is a useful proxy to estimate m6Anet performance in other cell lines.

**They analyzed or trained models with in vitro transcribed (IVT) from HeLA and E coli. The IVT for other cell lines can be used to assess false positives of calling RNA modifications since IVT samples should not have any modifications. This is another important point to address in light of the concerns above on over-calling “positive” sites.**

This is a great suggestion. We performed this analysis and found that Nanopolish finds over three times more false positive sites (143) than Uncalled4 (37), despite the fact

that Uncalled4 outputs 11% more sites overall. This result is much better than we had expected, and we are very appreciative of the suggestion.

#### **Remarks on code availability**

**More documentation is necessary to show how to use the software. For example, what is a pore model (the collection of preset models) and how to specify a custom pore model in a TSV file.**

Thank you for your suggestions. We have added a script to convert Uncalled4 pore models from their compressed numpy format to a more interpretable TSV format, and provided a better description of how they are defined.

~~~~~

Reviewer #3:

Remarks to the Author:

The authors, Sam Kovaka et. al. described a new nanopore signal alignment tool called, Uncalled4. This tool is designed to align the raw signal data from nanopore sequencers to a reference sequence, which is a fundamental step for any analysis using the nanopore raw signal. Uncalled4 is an impressive tool kit which represents a major improvement in nanopore signal alignment. The authors show that the dynamic time warping signal alignment method implemented within Uncalled4 increases the number of aligned segments and more accurately represents the nucleotide polymers that are measured by the nanopore sensors. The authors focus on how the signal alignment step can be used for a variety of analysis steps, such as inferring a pore model, and detecting nucleotide modifications. They clearly show how fundamental the signal alignment step is for modification detection, by testing the most widely used signal alignment tool, Nanopolish, and Uncalled4 as the data preprocessing step for RNA modification detection tools. When compared against annotated m6A sites, the modification detection software tools had higher scores when using Uncalled4 as the signal alignment tool instead of Nanopolish or Tombo. Despite showing an impressive tool, I have some points that I'd like the authors to clarify.

Thank you for your positive summary and comments.

Major points:

1. Figure 1 comments: Panel 1B: The Jaccard distance is defined in the text as "the fraction of raw samples that both methods aligned to the same position". However, in panel B, it appears that only one method is being displayed. How is the Jaccard distance being used in this panel? Panel C: legend description doesn't explain what the blue line (compare) is representing, more details here would help to better understanding the plot. Panel D: the label for the heatmap color bar appears to be cropped. Panel D: the red arrows are not labeled in the plot, nor the figure legend. I see what they are trying to depict, but I find them largely distracting and think that describing in the figure legend can provide the

necessary context for the other figure panels. Panel E: Including the Native and IVT legend in the plot would help for understanding the plot immediately.

Originally **Fig. 1b** displayed a comparison between Uncalled4 and *ref-moves*, but this was not obvious and didn't fit the structure of the results section. To address this, we have revised and rearranged **Fig. 1**, replacing the comparison statistics in panel **b** with the basic alignment layers (*current*, *current_sd*, and *dwell*), and also swapping the pore model figures with those in **Fig. 2** to better introduce pore model composition. We also remade all alignment panels to use the example *E. coli* rRNA dataset provided on GitHub, and display the same region in each panel.

2. In this sentence: "Nanopolish has slightly lower average Jaccard distance in r9.4.1 DNA and RNA, with less than 50% overlap on average for all aligners and less than 2% difference between Uncalled4 and Nanopolish" it appears that the Jaccard distance can be calculated individually for each tool, but in the text the Jaccard distance is defined as a comparative method. I am not super familiar with the Jaccard distance, but it is my understanding that it is a comparative statistic. Can the text be clarified what sets are being compared to calculate the Jaccard distance here and in other parts of the manuscript, such as in figure 1B?

Thank you for your suggestions. We have substantially rearranged **Fig. 1** and the opening results section, which we hope clarifies that distance can either be computed between two signal alignment methods, or between one method and the *ref-moves* which guided Uncalled4 alignment.

3. Do the authors expect the file size and computation time to scale linearly as data sets increase by 4000 reads?

We added **Supplemental Figure 4** to show that file size grows linearly for both Uncalled4 BAMs and the "eventalign" format. We also added a comparison to gzip-compressed eventalign, which is the compression algorithm used by the BAM format.

4. The color coding in Figure 4 e&f is not entirely clear. The negative values (green) suggest that these are m6A counts found by m6A net using Nanopolish as the signal alignment method, but not by m6Anet using Uncalled4 as the signal aligner. But does this example mean that no m6A sites were identified by m6Anet using Uncalled4 or simply X many more using Nanopolish?

We clarified the figure caption to explain we are displaying the difference in aggregate per-gene m6A count across all samples, so the "zero" values indicate the same number was found by both aligners.

5. In describing the dynamic time warping algorithm, much care was given to the alignment space banding, but a description of the distance equation was not included in the methods section.

We have added the DTW cost function to the methods section.

6. The two implemented data scaling methods are not entirely clear, what is the difference between reference-guided method-of-moments scaling and model-guided methods-of-moments scaling?

We expanded our description of the normalization algorithm to explain each iteration and how it improves normalization, including an equation for the “method-of-moments”.

Minor points:

1. The last sentence of first paragraph in the introduction is a little unclear what the authors mean by “although only a few have been detected thus far”. It seems at odds with the first half of the sentence that there are 150 known RNA modifications. Do the authors mean instead that only a few have been detected using nanopore technology?

“Over 150 known RNA modifications are known to exist, although only a few have been detected thus far, with varying accuracy [10].”

We updated the text to clarify “only a few can be comprehensively detected at the single-nucleotide level”.

2. I think that the right parenthesis should be moved in this sentence to only include the acronym “such as Dynamic Time Warping (DTW, used by Tombo) or Hidden Markov Models (HMMs, used by Nanopolish)” should be “such as Dynamic Time Warping (DTW), as used by Tombo or Hidden Markov Models (HMMs), as used by Nanopolish”.

Good point, this has been fixed.

3. Figure 1 panels B and C, is there a reason that they are different regions of the E. coli 16s rRNA?

We remade the relevant **Fig. 1** panels to display the same regions.

4. Throughout the text the authors use both 5mCpG and 5mC. I think it would be better to pick one naming convention where possible. The authors use 5mCpG six times and 5mC seven times.

We have replaced most instances with “5mCpG”, except in cases where we explicitly reference all-context 5mC or potentially unmodified CpG sites.

5. Table 1: It is my understanding that table legends should appear above the table instead of below like with figures.

We have moved the legend above the table

6. Supplemental figure 3e would help to include the color coding within the plot instead of leaving the color coding just to the figure legend.

This figure was moved to **Supplemental Figure 9** with an updated legend.

7. Supplemental figure 4 a&b would benefit from including the central kmer base color coding within at least one of these two plots.

We have added the legend, and this figure was moved to **Supplemental Figure 1** to introduce the RNA004 model earlier.

8. Supplemental figure 10 appears to have a legend and figure title mismatch. Also, the black dots in the plot are not described anywhere.

This figure has been remade, the caption corrected, and was relabeled **Supplemental Figure 15**. The black dots had represented the point where the F1 was maximized, but we didn't comment on it so we removed them.

Remarks on code availability

I made a fresh install of gcc and a clean conda virtual environment with python version 3.8, and tried to install uncalled4 using the github method. I was unable to install uncalled4. Here is hopefully a useful error code for the authors:

In file included from src/cpp/config.cpp:1:

In file included from ./src/cpp/config.hpp:30:

In file included from ./src/cpp/dtw.hpp:8:

In file included from ./src/cpp/seq.hpp:32:

In file included from ./src/cpp/pore_model.hpp:37:

./src/cpp/intervals.hpp:167:9: error: 'auto' deduced as

'std::vector<>::const_iterator' (aka '__wrap_iter<const Interval<long long> *>') in declaration of 'a' and deduced as 'std::vector<>::iterator' (aka '__wrap_iter *>') in declaration of 'b'

```
auto a = coords.begin(),
```

```
^ ~~~~~
```

./src/cpp/aln.hpp:175:25: note: in instantiation of member function 'IntervalIndex<long long>::intersect' requested here

```
index = a.index.intersect(b.index);
```

```
^
```

26 warnings and 6 errors generated.

error: command '/usr/bin/clang' failed with exit code 1

[end of output]

note: This error originates from a subprocess, and is likely not a problem with pip.

ERROR: Failed building wheel for uncalled4

Running setup.py clean for uncalled4

Failed to build uncalled4

ERROR: Could not build wheels for uncalled4, which is required to install pyproject.toml-based projects

However, independent of this review, one of my students was able to install uncalled4 on both his local machine and on the HPC. He has begun testing uncalled4 in our RNA modification detection framework. The code executed as expected in his hands.

The github documentation is clear (aside from the normalization comment mentioned above). The code is very readable, but there are several areas where the code was commented out, likely because those lines are no longer necessary. Also, it was a little difficult to follow the logic on how the different subcommands called the different functions when I was tried to track down the code for the convert subcommand.

I believe the error reported above was a compiler version error, and we have updated the documentation to specify a minimum GCC compiler requirement. We have also removed commented out and other unused code blocks, and have fixed several GitHub issues reported since the manuscript was submitted. We have also developed a pip package to further simplify installation and enhance usability (`pip install uncalled4`)

Reviewer #1 (Remarks to the Author):

The authors have addressed my comments. I don't have further comments

Reviewer #1 (Remarks on code availability):

the code is well documented and of high quality

Thank you for your comments.

Reviewer #2 (Remarks to the Author):

The authors have made an effort to address my previous comments. The additional analysis on IVT is especially interesting (and I do think it is very important). The use of additional "gold standard" other than or in addition to m6A-atlas is also important to assess the performance of the new approach. I think `uncalled4` could be a very useful tool to the community doing signal-level analysis of nanopore data and recommend the publication of this manuscript.

Reviewer #2 (Remarks on code availability):

It would be better if they have some example output (including figures from dotplot and trackplot), possibly in the `example/` folder, so that users know what are expected and can compare results to ensure the correct execution of the tool.

Thank you for this suggestion, we have added a few example outputs to the GitHub "example" directory as you suggested

Reviewer #3 (Remarks to the Author):

The authors of `uncalled4` have made many improvements to the manuscript and github pages since their initial submission. The added clarity on how the different pore models could be trained, and demonstrating the utility of a custom pore model for BrdU detection was a beneficial addition. Furthermore, rearranging figure 1 greatly improved the clarity on how the `uncalled4` DTW alignment compared with the move table and Nanopolish alignments. Additionally, the expanded methods section is much clearer. Splitting the RNA modification detection section made the logic and analysis easier to follow. In this updated manuscript, the authors have addressed all my previous comments, but I still have some minor comments about figure 4.

Thank you for your comments and suggestions

In figure 4, the authors use m6ACE-seq [<http://dx.doi.org/10.1038/s41587-021-00949-w>] as their ground truth set for panel A, but then use the GLORI as their ground truth set for panels B and C. As all the data in Figure 4 is from HEK293 cells, it seems like it would be more appropriate to use the same ground truth set for AUPRC calculations. Given that the AUPRC values in figure 4a range between 0.08 and 0.35, yet in figure 4b they range between 0.578 and 0.744, the m6ACE-seq seem ill suited as a ground truth set for m6A detection.

We kept the m6ACE-seq ground truth for the comparative results because those sites are filtered to only include METTL3-sensitive sites by comparing wildtype and METTL3 knockout samples. The GLORI results include all m6A sites, including those which aren't expected to be affected by the METTL3 knockout. The GLORI paper did include results from HeLa and HEK239T METTL3 knockdown experiments, however the supplemental data aren't sufficiently described to determine how the knockdown cells correspond to the full knockout experiments that we're comparing to. Furthermore, we believe that our comparisons between the GLORI results and the m6A-Atlas demonstrate that the relative performance of Uncalled4 and other aligners remains the same even when a less comprehensive ground truth is used, so we think the m6ACE-seq results are sufficient in this case.

The claim about gene-level modification calls being more robust than transcript-level could simply be explained by increased power from more observations for the statistical tests at each reference position than anything to do with ambiguous mapping to transcripts. And the AUPRC values for the two uncalled4 based alignments (the only tool that is appropriate for gene-level alignments) show only a modest improvement when switching to gene-level modification detection. Have the authors tried gene-level modification detection after downsampling to the average number of reads aligned to individual transcripts?

Thank you for highlighting this point. The main source of reduced performance for transcriptome alignment is indeed most likely reduced coverage for multi-isoform genes. Your suggested experiment may demonstrate that, however, it could unfairly disadvantage the gene-level results since the far greater number of transcripts increases the odds that one transcript will score high enough by chance. We have added a sentence to clarify these points in Supplemental Note 6.

Furthermore, it would make comparisons between transcript-level and gene-level easier if the y-axis ranges were identical for the All context and the DRACH context columns.

This would be a straightforward change, however we don't think it is appropriate given the nature of precision-recall curves between datasets with different background compositions. Unlike AUROC, AUPRC scores are not directly comparable when the number of candidate and positive sites are unequal, since it is much harder to have higher precision when a smaller fraction of sites are positive. The "All-Context" measurements include all DRACH sites in addition to all other adenines, and since that's a known property of AUPRC, it wouldn't be fair to suggest that the All-Context performance is globally worse.

The claim "We also computed KS statistics using uncorrected ref-moves and found it performed worse than KS statistics using any other alignment method" was not supported by any quantitative evidence in the prose, nor any figure or supplemental figure in the manuscript.

This is shown in Supplemental Figure 12. This claim has been moved to Supplemental Note 1, where we also discuss current MAD of uncorrected ref-moves, as it no longer fits within the shortened main text.

Reviewer #3 (Remarks on code availability):

The code base is organized really well and the improvements to the code since the initial submission have made it much more clear and easy to understand how the different parts work.

Thank you for your comment.